# Cannabinoid-induced actomyosin contractility shapes neuronal morphology and growth

Alexandre B Roland[1,2†], Ana Ricobaraza[1†], Damien Carrel[1†‡], Benjamin M Jordan[3], Felix Rico[4], Anne Simon[1], Marie Humbert-Claude[1], Jeremy Ferrier[1], Maureen H McFadden[1], Simon Scheuring[4], Zsolt Lenkei[1*]

[1]Brain Plasticity Unit, ESPCI-ParisTech, CNRS UMR8249, Paris, France; [2]FAS Center for Systems Biology, Harvard University, Cambridge, United States; [3]Department of Organismic and Evolutionary Biology, Harvard University, Cambridge, United States; [4]U1006 INSERM, Aix-Marseille Université, Parc Scientifique et Technologique de Luminy, Marseille, France

**Abstract** Endocannabinoids are recently recognized regulators of brain development, but molecular effectors downstream of type-1 cannabinoid receptor (CB1R)-activation remain incompletely understood. We report atypical coupling of neuronal CB1Rs, after activation by endo- or exocannabinoids such as the marijuana component $\Delta^9$-tetrahydrocannabinol, to heterotrimeric $G_{12}/G_{13}$ proteins that triggers rapid and reversible non-muscle myosin II (NM II) dependent contraction of the actomyosin cytoskeleton, through a Rho-GTPase and Rho-associated kinase (ROCK). This induces rapid neuronal remodeling, such as retraction of neurites and axonal growth cones, elevated neuronal rigidity, and reshaping of somatodendritic morphology. Chronic pharmacological inhibition of NM II prevents cannabinoid-induced reduction of dendritic development in vitro and leads, similarly to blockade of endocannabinoid action, to excessive growth of corticofugal axons into the sub-ventricular zone in vivo. Our results suggest that CB1R can rapidly transform the neuronal cytoskeleton through actomyosin contractility, resulting in cellular remodeling events ultimately able to affect the brain architecture and wiring.

*For correspondence: zsolt.
lenkei@espci.fr

†These authors contributed
equally to this work

Present address: ‡Université Paris
Descartes, Sorbonne Paris Cité,
CNRS UMR8250, Paris, France

**Competing interests:** The
authors declare that no
competing interests exist.

**Reviewing editor**: Franck Polleux,
Columbia University, United States

## Introduction

The endocannabinoid (eCB) system is emerging as an important regulator of brain wiring during development with a variety of functions, ranging from lineage segregation of stem cells to refinement of synaptic functions in complex neuronal networks (*Williams et al., 2003*; *Berghuis et al., 2007*; *Harkany et al., 2008*; *Mulder et al., 2008*; *Vitalis et al., 2008*; *Watson et al., 2008*; *Wu et al., 2010*). In both the embryonic and adult brains, eCB action is predominantly mediated by CB1 cannabinoid receptors (CB1Rs), which is one of the most highly expressed neuronal G-protein-coupled receptors (GPCRs), known to couple to $G_{i/o}$ heterotrimeric proteins (*Howlett, 2005*), but the molecular mechanisms by which CB1R shapes developing neurons remain mostly unknown. The exact role of eCBs in shaping the neuronal architecture is also under debate, since several reports indicate neurite retraction, while others found the induction of neurite outgrowth following CB1R activation (review in *Gaffuri et al., 2012*). Likewise, currently it is difficult to reconcile the locally repulsive effects of eCBs, reported at axonal growth cones (*Berghuis et al., 2007*; *Argaw et al., 2011*), and their role of mediating efficient directional axonal growth and shaping well-fasciculated axonal tracts (*Mulder et al., 2008*; *Vitalis et al., 2008*; *Watson et al., 2008*).

During neuronal development, an elaborate balance of positive and negative regulators is necessary to establish precise neuronal structure. This structure is stabilized by the cytoskeleton, which, similar

**eLife digest** Our brains are full of cells called neurons, which are connected to each other in complex networks that send messages around the brain. The way the neurons connect to each other, known as brain wiring, differs widely between individuals. Moreover, our brain wiring changes in response to our environment and experiences throughout our lives, from developing embryo to old age.

One way this happens is through the action of chemicals called cannabinoids. Produced naturally in the body, cannabinoids are also found in the popular recreational drug cannabis that is increasingly being used in medicine to treat chronic pain and other conditions. However, cannabis misuse can have negative side effects on the brain leading to memory loss and mental illness, especially in young people.

Cannabinoids can be detected by a group of proteins called cannabinoid receptors, but it is not clear how this leads to changes in brain wiring. Roland et al. now show that detection of cannabinoids by a type-1 cannabinoid receptor triggers a series of events that change how neurons grow and connect with each other.

Detection of the cannabinoid by the receptor leads to the activation of an enzyme called ROCK. This, in turn, activates a motor protein called non-muscle myosin II that inhibits the growth of neurons. Roland et al. suggest that this prevents the neurons from reaching their neighbors and forming new connections. Investigating how this works in individuals with medical conditions that alter brain function could help inform us how cannabis could be used more safely.

to non-neuronal cells, is composed of two major polymers, the highly plastic filamentous-actin (F-actin) and the more stable microtubule (MT) networks. Actin filaments are often cross-linked to a molecular motor protein, the non-muscle myosin II (NM II), whose contractile properties further endow the actomyosin network with highly dynamic control of cell behavior and architecture (*Vicente-Manzanares et al., 2009*). The cytoskeleton is mainly regulated by Rho-like GTPases that control a wide variety of effector mechanisms such as actin polymerization and branching, actomyosin contractility, focal adhesions, microtubule dynamics, and membrane transport (*Kaibuchi et al., 1999*; *Etienne-Manneville and Hall, 2002*; *Hall and Lalli, 2010*). Downstream protein kinases such as the Rho-associated, coiled coil-containing kinase (ROCK) are the key activator proteins of these convergent-signaling pathways. Interestingly, ROCK is associated with particular CB1R-induced phenotypes. In CB1R-over-expressing B103 cells, the endocannabinoid anandamide induces cell rounding via ROCK (*Ishii and Chun, 2002*), and CB1R activation results in RhoA- and ROCK-dependent repulsion of growth cones of cultured hippocampal neurons (*Berghuis et al., 2007*), but neither the coupling mechanism of CB1R to ROCK nor the cytoskeletal targets downstream of CB1R-activated ROCK are identified yet. Since Rho-activated effectors operate over a large range of spatial and temporal scales, understanding of eCB-mediated structural plasticity requires the identification of the precise spatial and temporal dynamics of CB1R-mediated cytoskeletal modifications.

In this study, by using highly resolved live imaging approaches, we report that CB1R-activation rapidly and reversibly contracts the neuronal actomyosin cytoskeleton through an unusual coupling to $G_{12}/G_{13}$ proteins that produce Rho- and ROCK-mediated NM II activation. In addition, we show that chronic CB1R-mediated activation of actomyosin contractility may mediate lasting changes in neuronal and cerebral morphology.

## Results

### CB1R-activation results in rapid retraction of actin-rich growth cones

In order to investigate the spatio-temporal dynamics of cannabinoid-induced cytoskeletal modifications, we have established a sensitive, specific, and highly accessible experimental assay system to study neuronal remodeling downstream of CB1R activation. We have visualized highly dynamic neuronal growth cones in cultured hippocampal neurons, where the activation of endogenous CB1Rs results in repulsion (*Berghuis et al., 2007*), by labeling endogenous F-actin with fluorescent LifeAct. This actin-binding peptide allows observation of the dynamic actin network without perturbing natural reorganization kinetics (*Riedl et al., 2008*).

Time-lapse microscopy of live neurons, expressing Flag-CB1R-eGFP and LifeAct-mCherry, showed numerous F-actin-rich dynamic growth cones (*Figure 1A*) advancing at individually variable velocities (*Figure 1A,B*), but yielding a fairly constant mean growth rate of 20–30 µm/hr (*Figure 1D*). In addition

to growth cones, axonal F-actin was also present in filopodia and in isolated patches on the shaft of the distal axonal region (*Figure 1—figure supplement 1*). Strikingly, bath application of 100 nM WIN 55,212-2 (WIN), a synthetic cannabinoid agonist, led to a rapid retraction of the F-actin-rich domain (*Figure 1A*), with mean retraction amplitude of 62.2 µm ± 5.2 (*Figure 1C–E*). Retraction was already detectable at 2 min after agonist exposure and typically reached a plateau between 10 and 20 min (*Figure 1C,D*). The morphology of retracted axons was characterized by an F-actin-rich retraction bulb (arrowheads on *Figure 1A* and *Figure 1—figure supplement 1*) and a thin membranous trailing remnant (open arrowheads on *Figure 1A* and *Figure 1—figure supplement 1*), the latter of which was not included in the length measurement. Pre-treatment with the CB1R selective antagonist/inverse agonist AM281 (AM) (1 µM) inhibited retraction (*Figure 1D,E*).

Further pharmacological characterization showed that several other chemically distinct CB1R agonists, the endocannabinoid 2-arachidonoylglycerol (2-AG) (1 µM), the principal psychoactive marijuana constituent $\Delta^9$-tetrahydrocannabinol ($\Delta^9$-THC) (1 µM) and the synthetic agonists CP55,940 (100 nM) and HU-210 (100 nM) also produced significant retraction (*Figure 1E*). The retraction was saturable and concentration-dependent with a half-maximal effective concentration ($EC_{50}$) value of around 20 nM for WIN (*Figure 1F*). When treatment with 25 nM WIN was followed by ligand-free wash-out, growth cone progression resumed normally showing the reversibility of cannabinoid-induced growth cone retraction (*Figure 1G*). Finally, this retraction was not a result of CB1R over-expression since treatment with 100 nM WIN or 1 µM 2-AG induced significant retraction with similar kinetics in neurons transfected only with LifeAct-mCherry (*Figure 1H–I*). However, the mean amplitude of retraction was lower and responses were more variable than in Flag-CB1R-eGFP-expressing neurons (compare *Figure 1C,D* with *Figure 1H,I*), as expected in a heterogeneous neuronal population expressing endogenous CB1Rs at highly variable levels (*Leterrier et al., 2006*). In addition, growth cone advance rapidly resumed even in the continued presence of 100 nM WIN (*Figure 1H*).

In conclusion, our results show that cannabinoids trigger a rapid, saturable, and reversible retraction of actin-rich growth cones downstream of both endogenous and overexpressed CB1Rs.

## $G_{12}/G_{13}$ heterotrimeric proteins, Rho GTPase, ROCK, myosin II, and F-actin microfilaments mediate CB1R-induced rapid growth cone retraction

First, we investigated which cytoskeletal elements act downstream of CB1Rs to induce rapid growth cone retraction. We expressed, in addition to LifeAct-mCherry, a GFP-tagged version of End-binding protein 3 (EB3-eGFP), which binds to endogenous microtubule (MT) plus ends without changing MT growth parameters and thus allows the visualization of MT structure and dynamics (*Stepanova et al., 2003*). Indeed, MTs in the entire neuron were labeled in green, with many bright comet-like fluorescent dashes in all the neuronal compartments, moving randomly in the cell body and directionally in axons and distal dendrites, representing dynamic MT plus ends (*Stepanova et al., 2003*). During 100 nM WIN-induced retraction the dynamics of the two main cytoskeletal polymers, F-actin and MTs, was remarkably different (*Figure 2A*). A significant portion of F-actin redistributed in the first 2–4 min after stimulation from its original location in growth cones into a more homogenous cable-like pattern on the distal axonal shaft (*Figure 2B* and *Figure 2—figure supplement 1*). In contrast, MTs bent during the same time frame forming periodic local loops (*Figure 2A,A′,B* and *Video 1*) before finally consolidating into a homogenously labeled retraction bulb. The F-actin cables (bundles of F-actin filaments, which are not separately resolved here by diffraction-limited microscopy), often co-localized with regions displaying periodic bends in MTs (*Figure 2B′*), suggesting that an F-actin-related force pulls strongly enough to bend MTs. This effect was not the result of the over-expression of the cytoskeletal markers EB3-eGFP or LifeAct-mCherry, since we could observe similar periodic MT bends, detected by post hoc immunohistochemistry, in neurons not expressing these markers (*Figure 2—figure supplement 2*).

To investigate the requirement for polymerized actin microfilaments and MTs in these retractions, we depolymerized MTs with nocodazole (10 µM) and F-actin with cytochalasin D (1 µM) (*Forscher and Smith, 1988*). Nocodazole pre-treatment stopped growth cone advance but WIN still induced significant retraction (*Figure 2C,D*, *Figure 3E* and *Video 2*). In contrast, cytochalasin D inhibited both growth cone advance and WIN-induced retraction (*Figure 2E,F*, *Figure 3E* and *Video 3*) showing that while the presence of both F-actin and MTs is necessary for growth cone advance, as reported previously (*Dent et al., 2003*), only F-actin is necessary for CB1R-induced retraction.

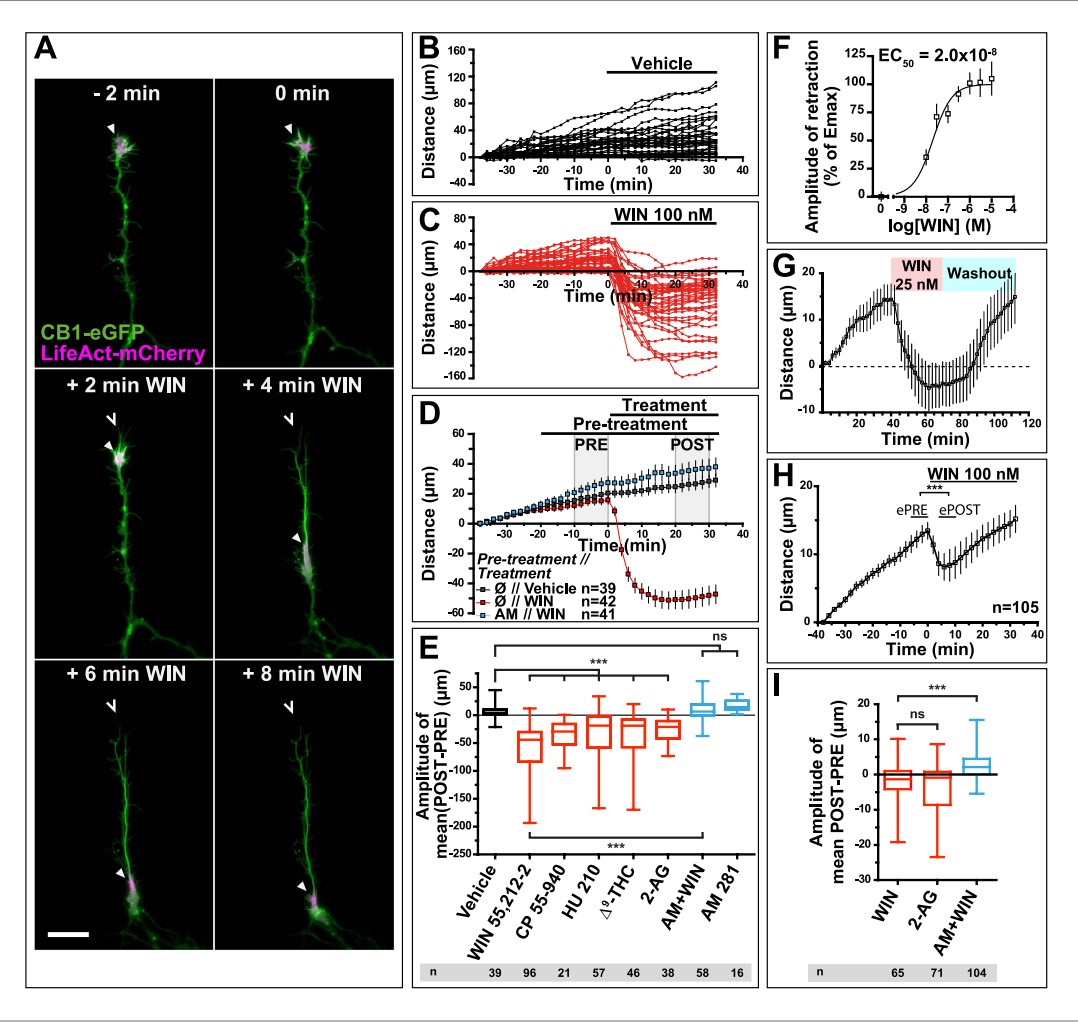

**Figure 1**. CB1R activation induces retraction of actin-rich growth cones. Cultured DIV8 hippocampal neurons co-expressing Flag-CB1R-eGFP and LifeAct-mCherry on (**A**–**G**) and LifeAct-mCherry only on (**H** and **I**). (**A**) Treatment with CB1R agonist WIN55,212-2 (WIN, 100 nM, added at 0 min) induces rapid retraction of the F-actin-rich domain (arrowheads). Open arrowheads: growth cone position at 0 min. (**B**) Progression of individual growth cones in control conditions. (**C**) WIN-induced retraction of individual growth cones. (**D**) Mean values of growth cone progression in control condition or after treatment with WIN with or without pre-treatment with the CB1R-specific antagonist AM281 (AM, 1 µM). WIN-induced growth cone retraction is effectively abolished by AM. (**E**) Amplitudes of growth cone retraction induced by different exo- and endocannabinoids, calculated as the net difference of mean growth cone position in the pre-treatment (PRE on **D**) and post-treatment (POST on **D**) time intervals from at least three independent experiments. (**F**) Concentration-response curve of WIN-induced retraction, 9 to 27 neurons per concentration from two independent experiments expressed as percentage of maximal retraction, Emax = 52.2 µm. (**G**) WIN-induced retraction (25 nM at 40 min) is fully reversible after WIN-washout (at 70 min), n = 9. (**H**) Mean values of growth cone retraction downstream of endogenous CB1R activation, from four pooled independent experiments, outliers were removed in accordance with the Grubb's test. (**I**) Amplitudes of growth cone retraction downstream of endogenous CB1R activation after treatment with WIN (100 nM), 2-AG (1 µM), or with WIN (100 nM) after pre-treatment with the CB1R-specific antagonist AM281 (AM, 1 µM). WIN-induced growth cone retraction is effectively abolished by AM. Values in **D**, **F**, **G**, and **H** are mean ± SEM; values in **E** and **I** are presented as boxplots; n.s = p > 0.05, ***p < 0.001, calculated using Kruskal–Wallis one-way ANOVA followed by Dunn's post-tests on (**E** and **I**) and paired t-test on (**H**). Scale bar: 20 µm.
The following figure supplement is available for figure 1:

**Figure supplement 1**. mCherry-LifeAct label (red channel) from *Figure 1A*. Scale bar: 20 µm.

A likely candidate for the generation of such rapid F-actin-related force, which is capable of bending microtubules, is non-muscle myosin II (NM II), an ATPase protein with actin cross-linking and contractile properties, which is activated by the phosphorylation of its regulatory light chain. The two main activators of NM II are myosin light chain kinase (MLCK) and ROCK, the latter being already known to participate in CB1R-induced cytoskeletal modifications (*Ishii and Chun, 2002*; *Berghuis et al., 2007*). This raises the possibility that ROCK- and/or MLCK-induced NM II contractility is responsible for the force-generation reported above. In order to directly investigate the implication of NM II, we pre-incubated neurons, for 20 min before WIN stimulation, with the highly selective NM II ATPase inhibitor blebbistatin (25 µM) that blocks NM II in an actin-detached state without perturbing F-actin polymerization (*Kovacs et al., 2004*). Blebbistatin pre-treatment induced substantial morphological changes of the growth cone, which continued to move forward in a rather disorganized fashion (*Figure 2G* and *Video 4*), typically transforming the growth cone lamellipodia into several dynamically advancing filopodia, as reported previously (*Rosner et al., 2007*). Remarkably, blebbistatin completely abolished WIN-mediated retraction of these dynamically advancing F-actin-rich structures (*Figure 2G,H*, *Figure 3E* and *Video 4*), suggesting that the main force-generating factor downstream of CB1R activation is actomyosin contractility. This inhibitory effect of blebbistatin was concentration dependent with half-maximal value of inhibition ($EC_{50}$) of 116 nM (*Figure 2—figure supplement 3*). Immunocytochemical analysis of WIN-treated F-actin-rich growth cones at 2 min after the addition of WIN strikingly showed rapid and strong up-regulation of myosin light chain phosphorylation in the distal axon, adjacent to the F-actin-rich growth cone (*Figure 3A–C* and *Figure 3—figure supplement 1*), at the right place for the subsequent NMII-dependent contraction, both in neurons transfected only with LifeAct-mCherry (*Figure 3A–C*) and with Flag-CB1R-eCFP and LifeAct-mCherry (*Figure 3C*).

Next, we investigated the mechanism coupling CB1R to the ROCK/NM II pathway. First, we showed that NMII-dependent growth cone contraction is not a result of CB1R over-expression, since treatment with blebbistatin (25 µM) or the ROCK inhibitor Y-27632 (10 µM) (*Figure 3D*) significantly inhibited endogenous CB1R-induced retraction of growth cones, previously presented on *Figure 1I*, in neurons transfected only with LifeAct-mCherry and EB3-eGFP. Then we used neurons expressing Flag-CB1R-eCFP, LifeAct-mCherry, and EB3-eGFP, our high-throughput experimental read-out, to characterize in detail the molecular mechanism of CB1R-induced actomyosin contractility. The amplitude of WIN-mediated retraction was significantly reduced by pre-treatment with the Rho inhibitor C3 transferase (1 µg/ml, *Figure 3—figure supplement 2*), the ROCK inhibitor Y-27632 (10 µM) (*Figure 3E*), but not by the MLCK-specific inhibitor ML-7 (30 µM) (*Figure 3E*). Treatment with the inactive (R)-(+)-blebbistatin (25 µM) stereoisomer was ineffective (data not shown). The implication of neuronal NM II was further confirmed by siRNA knock-down of endogenous NM IIA and NM IIB (*Miserey-Lenkei et al., 2010*), which resulted in significant reduction of WIN-mediated contractility as compared to control (anti-luciferase) siRNA (*Figure 3F*).

Next, we investigated which heterotrimeric G-protein family couples CB1Rs to Rho activation. Notably, treatment with pertussis toxin (100 ng/µl), a specific inhibitor of $G_{i/o}$ heterotrimeric proteins, which are generally considered as the main signaling pathway of CB1Rs (*Howlett, 2005*), did not decrease significantly cannabinoid-induced growth cone retraction (*Figure 3E*), similarly to a previously reported finding for ROCK-mediated induced cell rounding after anandamide treatment (*Ishii and Chun, 2002*). Another family of heterotrimeric G-proteins, $G_{12}/G_{13}$, may mediate rapid growth cone collapse, neurite retraction, and cell rounding in neuronal cell lines in response to certain GPCR agonists such as lysophosphatidic acid (LPA) (*Katoh et al., 1998*; *Kranenburg et al., 1999*). Therefore, we inactivated endogenous $G_{12}/G_{13}$ proteins in our hippocampal neuronal cultures by using two pools of 4 different siRNAs directed against rat $G_{12}$- or $G_{13}$-alpha proteins, respectively. Used separately, neither pool decreased WIN-induced growth cone retraction as compared to control (anti-luciferase) siRNA (*Figure 3F*). However, when we combined together 2 siRNAs of each pool, each resulting mixed pools efficiently inhibited WIN-mediated contractility (*Figure 3F*). These results show that the presence of either $G_{12}$ or $G_{13}$ is necessary and sufficient for CB1R-induced actomyosin contraction.

Finally, to verify that CB1R-induced retraction is not an artifact of altered adhesion properties of growth cones in vitro, we co-transfected Flag-CB1R-eCFP, EB3-eGFP, and LifeAct-mCherry into embryonic rat brains using in utero electroporation at embryonic day 16 (E16). In organotypic slices prepared from the offspring between postnatal day 4 and 6 (P4–P6), numerous corticofugal F-actin-rich growth cones from layer II–III pyramidal neurons could be visualized by video microscopy at 48 hr

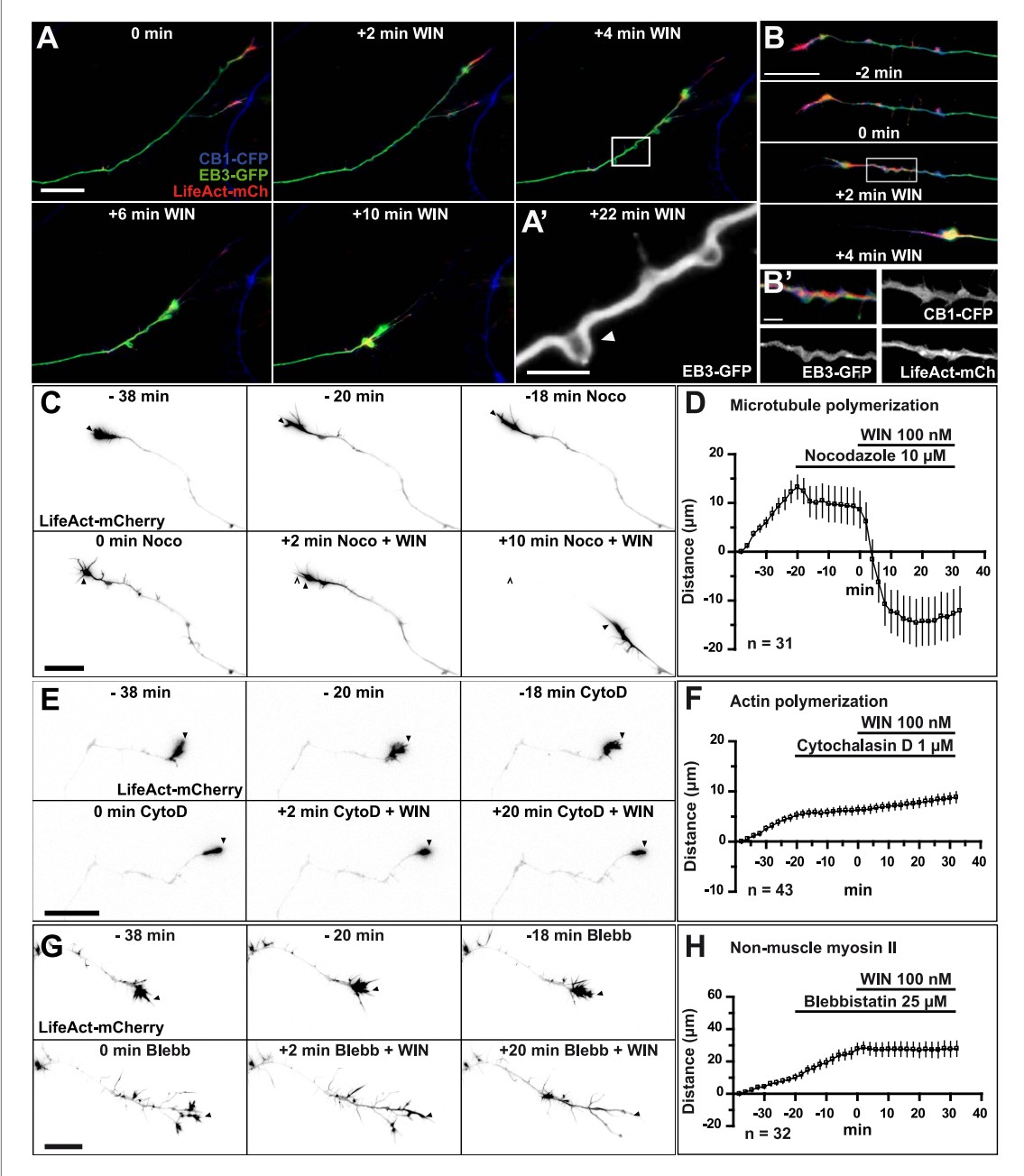

**Figure 2**. CB1R-induced retraction is mediated by non-muscle myosin II dependent actomyosin contraction. Cultured hippocampal neurons co-expressing Flag-CB1R-eCFP, LifeAct-mCherry, and EB3-eGFP at DIV6 were treated by WIN (100 nM) at 0 min. (**A**) Microtubules (MT) bend and form small loops (arrowhead on **A'**) in the first 4 min (**B**) F-actin is reorganized from the growth cone tips and isolated patches to homogenous cable-like distribution in distal axonal shaft. (**C–H**) Pre-treatment with: (**C** and **D**) MT polymerization inhibitor nocodazole (10 µM), (**E** and **F**) actin polymerization inhibitor cytochalasin D (1 µM), (**G** and **H**) Non-muscle myosin II-inhibitor blebbistatin (25 µM). Scale bars: 5 µm on (**A'**) and (**B'**), 20 µm elsewhere.

The following figure supplements are available for figure 2:

**Figure supplement 1**. Averaged F-actin relocalization in the distal 60 µm in growth cones in the first 4 min after WIN treatment in five randomly chosen neurons from *Figure 1C*.

**Figure supplement 2**. CB1R-induced periodic microtubule bends are not due to EB3-eGFP and LifeAct-mCherry expression.

**Figure supplement 3**. Concentration-response curve for the blebbistatin effect on the growth cone retraction essay after treatment with WIN (100 nM).

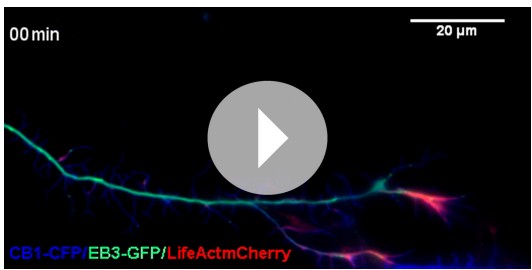

**Video 1**. CB1R activation induces retraction of actin-rich growth cones. Dynamic, F-actin-rich growth cone of a cultured hippocampal neuron co-expressing CB1R-eCFP, LifeAct-mCherry, and EB3-eGFP at DIV6 treated with 100 nM WIN at 10 min. Scale bar: 20 µm.

after slice preparation (*Figure 4A*). Application of 1 µM WIN resulted in significant retraction of growth cones (*Figure 4B,D* and *Video 5*) through activation of CB1R since this effect could be prevented by pre-treatment with 5 µM AM281 (*Figure 4D*). This retraction displayed slower kinetics ex vivo than in vitro (compare to *Figure 1*) probably due to limited diffusion of the highly hydrophobic WIN into the slice and/or into differences in adhesive and mechanistic properties within the organotypic brain slice. Previously, we have shown that at around P5, cortical projection neurons still express CB1R, albeit at lower levels than at birth (*Vitalis et al., 2008*), thus we have replicated these experiments by expressing only the cytoskeletal markers EB3-eGFP and LifeAct-mCherry. WIN-mediated activation of endogenous CB1Rs typically led to arrest or retraction of numerous growth cones (*Figure 4C,E*). The relatively mild averaged effect is probably due to the variable level of endogenous CB1R expression in these neurons. Importantly, pre-treatment with blebbistatin (25 µM) efficiently blocked this effect (*Figure 4E*).

In conclusion, we show that CB1R activation significantly reorganizes growth cones through MLCK/ROCK-mediated NM II activation. This large-scale actomyosin contractility ultimately leads to the remodeling of MT structure in the distal axonal segments.

### In the developing brain, both activation of endogenous CB1Rs and actomyosin contractility are required for path-finding of CB1R expressing corticofugal axons

In the embryonic brain, developing corticofugal axons express high levels of CB1Rs (*Figure 5B',B"*) (*Vitalis et al., 2008*). Genetic or pharmacological ablation of CB1Rs leads to axonal fasciculation deficits (*Mulder et al., 2008*; *Watson et al., 2008*). In order to investigate the importance of actomyosin contractility during the embryonic development of CB1R expressing axons, we inhibited in vivo the ATPase activity of NM II by in utero intra-cerebroventricular injection of rat embryos with blebbistatin (*Figure 5A*). Notably, 100% of blebbistatin-injected embryos survived and developed without apparent gross anatomical brain defects, suggesting that neuronal NM II can be safely targeted in vivo. In embryos treated between E15 and E17 with active (S)-(−)-blebbistatin (*Figure 5D,D´,G*), but not with the inactive (R)-(+) stereoisomer (*Figure 5C,C´,G*), Tuj1-expressing axons showed important targeting errors, by invading the sub-ventricular zone, from which CB1R-expressing corticofugal axons are usually excluded (*Figure 5B,G* and *Figure 5—figure supplement 1*). Such a representative CB1R-expressing Tuj1-positive axon invading the SVZ from an embryon treated with (S)-(−)-blebbistatin is shown on *Figure 5F*. Treatment with the CB1R-specific antagonist AM251 (1 mM) but not with its vehicle (DMSO 2.8%) led to similar developmental phenotype (*Figure 5E,E´,G*).

Together with our above findings showing that activation of endogenous CB1Rs in organotypic slices leads to NM II-dependent arrest or contraction of axonal growth cones, these results suggest that both activation of endogenous CB1Rs and actomyosin contractility are required for correct path finding of corticofugal axons.

### CB1R-induced rapid neuronal contraction results in cell rounding, neurite retraction, and increased cell stiffness in Neuro2A cells

Next, we asked whether the above reported CB1R-mediated effect on neuronal actomyosin contractility is restricted to growth cones, which are highly specialized mobile structures, or if we can also observe this phenomenon in other neuronal sub-compartments. The mouse neuroblastoma-derived Neuro2A cell line, a widely used model of neuronal physiology, presents simpler morphology than primary hippocampal neurons, enabling high-resolution quantitative measure of cellular structure and biomechanical characteristics. The Neuro2A cells grow neurites in culture, and we observed that F-actin accumulates in the shaft and extremity of these neurites as well as in highly dynamic filopodia and in patches of the cell cortex (*Figure 6A*). CB1R-eGFP showed a characteristic distribution between the plasma membrane and endosomes, as described previously in various non-polarized cell-types

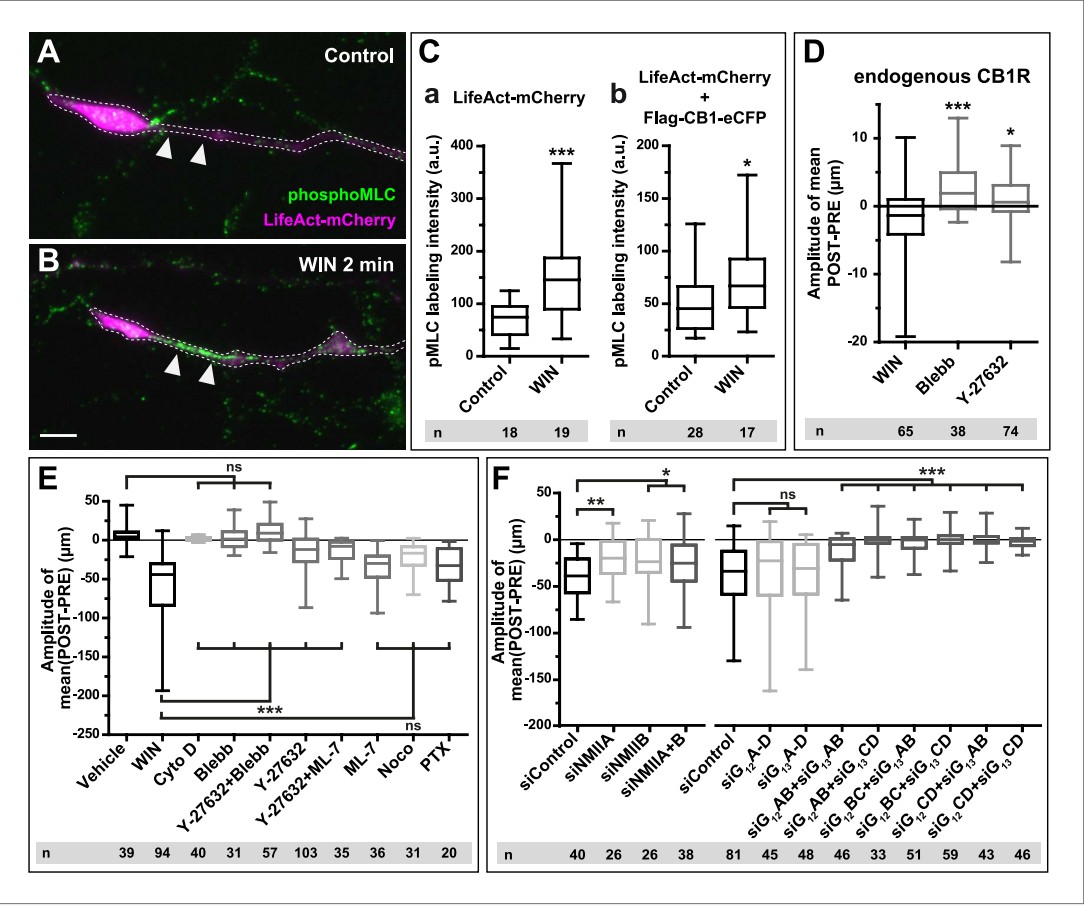

**Figure 3**. CB1Rs activate non-muscle myosin II through heterotrimeric $G_{12}/G_{13}$ proteins, Rho GTPase, and ROCK. Cultured hippocampal neurons at DIV6 co-expressing a combination of LifeAct-mCherry, Flag-CB1R-eCFP, and EB3-eGFP as indicated and treated by WIN (100 nM) at 0 min. (**A–B**) Representative LifeAct-mCherry expressing growth cones (delimited with a dotted line) at 2 min after treatment with vehicle (**A**) or WIN (100 nM, **B**), labeled with a phospho-Myosin Light Chain (phosphoMLC) antibody. Arrowheads show the distal axon adjacent to the F-actin-rich growth cone where WIN induces rapid and strong upregulation of myosin light chain phospho-rylation. (**C**) pMLC labeling intensity at the distal 50–60 µm of the axon, adjacent to the actin-rich growth cone, from neurons expressing LifeAct-mCherry (**A**) or co-expressing LifeAct-mCherry and Flag-CB1R-eCFP (**B**). The region-of-interest used to measure pMLC labeling intensity is delimited with a dotted line on a representative growth cone on *Figure 3—figure supplement 1*. (**D**) Amplitude of 100 nM WIN-induced growth cone retraction in neurons co-expressing LifeAct-mCherry and EB3-eGFP pre-treated with 25 µM blebbistatin or 10 µM Y-27632. (**E**) Amplitude of 100 nM WIN-induced growth cone retraction in neurons co-expressing LifeAct-mCherry, EB3-eGFP, and Flag-CB1R-eCFP pre-treated with: 1 µM cytochalasin D; 25 µM blebbistatin; 25 µM blebbistatin + 10 µM Y-27632; 10 µM Y-27632; 30 µM ML-7 + 10 µM Y-27632; 30 µM ML-7; 10 µM nocodazole; 100 ng/µl PTX. (**F**) Effect of siRNA-mediated knock-down of endogenous myosin IIA, IIB or of endogenous $G_{12}/G_{13}$ proteins on growth cone-retraction induced by 100 nM WIN in neurons co-expressing the three constructs, as compared to control (luciferase) siRNA. Results are pooled from at least two independent experiments, and outliers were removed in accordance with Grubb's test. Results in are expressed as boxplots. n.s p > 0.05; *p < 0.05; **p < 0.01; ***p < 0.001 calculated using Student's t-test on (**C**), Kruskal–Wallis one-way ANOVA followed by Dunn's post-tests on (**D**) and (**E**), and using one-way ANOVA followed by Newman–Keuls post-tests on (**F**). Scale bar: 10 µm.

The following figure supplements are available for figure 3:

**Figure supplement 1**. Another representative LifeAct-mCherry expressing growth cone 2 min after treatment with WIN (100 nM), labeled with the phosphoMLC antibody, similarly to *Figure 3B*.

**Figure supplement 2**. Amplitude of 100 nM WIN-induced growth cone retraction in neurons co-expressing LifeAct-mCherry, EB3-eGFP, and Flag-CB1R-eCFP with (C3T) or without (WIN) pre-treatement with 1 µg/ml C3T.

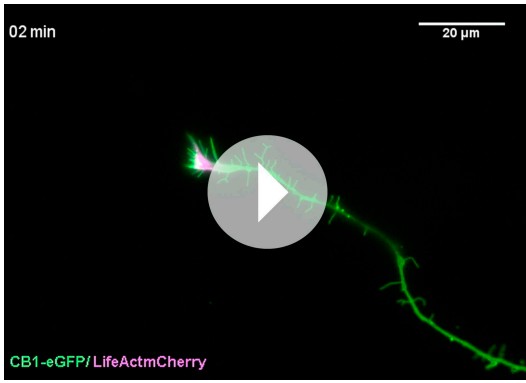

**Video 2**. Effect of microtubule depolymerization on CB1R-induced growth cone retraction. Dynamic, F-actin-rich growth cone of a cultured hippocampal neuron co-expressing Flag-CB1R-eGFP and LifeAct-mCherry at DIV6, pre-treated with 10 µM Nocodazole at 20 min before treatment with 100 nM WIN at 40 min. Scale bar: 20 µm.

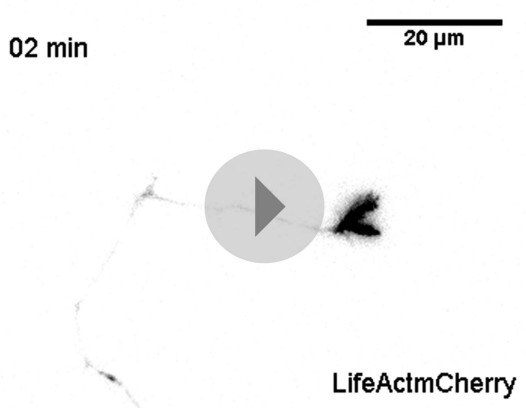

**Video 3**. Effect of actin depolymerization on CB1R-induced growth cone retraction. Dynamic, F-actin-rich growth cone of a cultured hippocampal neuron co-expressing CB1R-eCFP, LifeAct-mCherry, and EB3-eGFP at DIV6 pre-treated with 1 µM cytochalasin D at 20 min before treatment with 100 nM WIN at 40 min. Scale bar: 20 µm.

(*Leterrier et al., 2004*). Treatment with 100 nM WIN resulted in rapid rounding of the cell body and in retraction of neurites, leaving behind retraction-fiber-like remnants (*Figure 6A,B* and *Video 6*). F-actin was reorganized and accumulated at the end of the retraction bulb and under the plasma membrane of the cell body. This WIN-induced cell rounding could be blocked by blebbistatin treatment (25 µM) (*Figure 6B*), suggesting that the observed rapid morphological changes are due to a CB1R-induced general contraction of the actomyosin cell cortex, which is the association of plasma membrane lipids and the underlying actin filament network.

Comparable large-scale contraction of the actomyosin cortex was previously characterized in detail in cells entering division (*Théry and Bornens, 2008*), where a regulated balance between localized actomyosin-cortex-dependent surface tension and intracellular pressure allows dividing cells to control their volume, shape, and mechanical properties (*Stewart et al., 2010*). When combined, these two effects result in an overall increase of cell cortex rigidity or stiffness (*Stewart et al., 2010*), while local and temporary detachment of the plasma membrane from the actomyosin cortex results in characteristic blebbing (*Cunningham, 1995*; *Charras et al., 2008*). Marked cell rounding, F-actin reorganization, and the presence of retraction fibers suggested that an analogous intracellular mechanism might be implicated in the above-reported cannabinoid-induced reorganization of the Neuro2A cells. We have performed two experiments to investigate this possibility.

First, in order to directly measure putative contraction of the neuronal actomyosin cortex, we measured the cell cortex rigidity of isolated CB1R-expressing Neuro2A cells before and after cannabinoid treatment, by using atomic force microscopy (AFM). Averaged AFM measurements in force mode with a 1-µm spherical bead attached to the cantilever (*Figure 6C*) indicated that the stiffness (Young's modulus) of unstimulated individual cells was approximately 300 Pa, close to values reported in acutely isolated hippocampal glial cells and neurons (*Lu et al., 2006*). For this series of experiments, cells were grown on uncoated plastic, a highly adhesive substrate, in order to minimize the displacement of Neuro2A cells during the measurement. WIN stimulation led to an overall rapid and transient increase of cell stiffness at different locations on the same cell, with the exception of the trailing edge (*Figure 6C4*). As the contribution of the underlying coverslip was significant in neurites (compare ordinate scale of *Figure 6C1* with *Figure 6C2–4*), in the following experiments we have centered our force measurements on the cell bodies, corresponding to the positions 2 and 3 on *Figure 6C*. The important transient WIN-induced increase in cell stiffness was absent during incubation with vehicle solution and could be prevented by pre-treatment with blebbistatin (25 µM) (*Figure 6D*), showing that activation of NM II is necessary to induce the measured changes.

Next, in order to follow morphological changes at the plasma membrane in detail, high-resolution time-lapse image stacks of retracting WIN-treated Neuro2A cells were acquired, deconvoluted, and

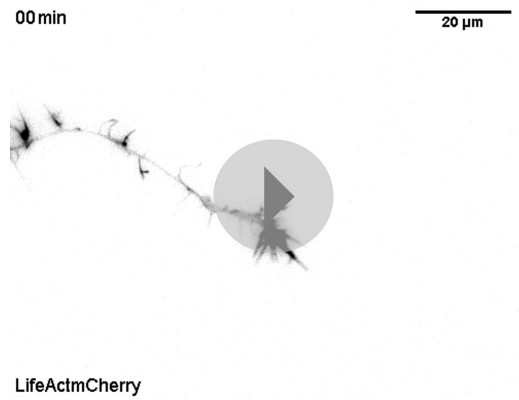

00 min

20 µm

LifeActmCherry

**Video 4**. Effect of NM II inhibition on CB1R-induced growth cone retraction. Dynamic, F-actin-rich growth cone of a cultured hippocampal neuron co-expressing Flag-CB1R-eGFP and LifeAct-mCherry at DIV6 pre-treated with 25 µM blebbistatin at 20 min before treatment with 100 nM WIN at 40 min. Only LifeAct-mCherry emission is visualized here. Scale bar: 20 µm.

reconstructed in three dimensions (*Figure 6E* and *Video 7*). Prior to remodeling, the shape of the cells suggested a large degree of reinforcement probably owing both to the intracellular actomyosin cortex directly beneath the plasma membrane, and to the attachment of the cell to its substrate. After CB1R agonist application, the cell changed shape drastically, acquiring a more spherical morphology. Moreover, we observed localized blebbing behavior of the cell membrane, starting at the early stages (~2 min) of the contraction and ceasing after 6–8 min (*Figure 6E* and *Video 7*).

In conclusion, our results show that CB1R activation leads to rapid and NM II-dependent neurite retraction and rounding of the cell body in Neuro2A cells, which is accompanied by formation of retraction fibers and by transient increase in cell stiffness and blebbing behavior. Collectively, these findings suggest that global CB1R activation results in large-scale contraction of the neuronal cytoskeleton, which is mechanistically similar to the molecular machinery engaged in mitotic cell rounding.

## Prolonged CB1R-mediated induction of actomyosin contraction reshapes somatodendritic morphology

Previously, we have reported that chronic in vitro activation of CB1Rs leads to significant inhibition of dendritic development in cultured hippocampal neurons, while genetic or pharmacological inhibition of CB1Rs leads to more numerous and longer dendrites (*Vitalis et al., 2008*). Similarly, genetic or pharmacological inhibition of CB1R leads to more complex somatodendritic morphology in septal cholinergic neurons (*Keimpema et al., 2013*). These data suggest that, in addition to axons, where CB1Rs are naturally targeted through transcytotic targeting (*Leterrier et al., 2006*), the transitory presence of CB1Rs on the somatodendritic membrane may allow efficient coupling to growth inhibitory signaling pathways. It was reported that increased NM II activity, through constitutively active MLCK or RhoA, decreases both the length and number of neurites and, consequently, delays or abolishes the development of neuronal polarity in cultured hippocampal neurons (*Kollins et al., 2009*). We thus studied whether a long-term effect of the above described rapid, CB1R-activation dependent and NM II-mediated contraction of the neuronal cytoskeleton could explain the negative regulatory effects of CB1R activation at a longer time scale (~24 hr).

First, we verified the presence of the rapid structural effects of CB1R activation in the somatodendritic region. Neurons expressing CB1R-eCFP, LifeAct-mCherry, and EB3-eGFP at DIV9 responded to 100 nM WIN with rapid morphological reorganization of the somatodendritic compartment, characterized by retraction of distal dendritic regions and broadening of the proximal portion of dendrites (*Figure 7A,A'* and *Video 8*). While the overall dynamics of EB3-eGFP comets was not apparently modified, the MTs in individual dendrites often displayed a characteristic bent morphology, parallel to the appearance of straight cable-like F-actin bundles (*Figure 7A,A'* and *Video 8*) suggesting the presence of a rapid CB1R-activation-dependent actomyosin contraction. Overnight treatment with WIN (100 nM) resulted in a significant decrease in the number of dendrites of developing hippocampal neurons, expressing Flag-CB1R-eGFP and the soluble cytoplasmic marker DsRed2 at DIV4 (*Figure 7B,C*), as reported previously (*Vitalis et al., 2008*). This effect was abolished in the presence of both Y-27632 (10 µM) or blebbistatin (25 µM) (*Figure 7B,C*). Notably, treatment with both inhibitors led to more developed dendrites also in control conditions, confirming previous reports on the constitutive inhibition of neurite development through ROCK and NM II (*Kollins et al., 2009*).

In conclusion, our results show that the chronic activation of CB1Rs reshapes somatodendritic morphology through enhancement of the naturally present ROCK- and NM II-mediated contractile tone of neurons.

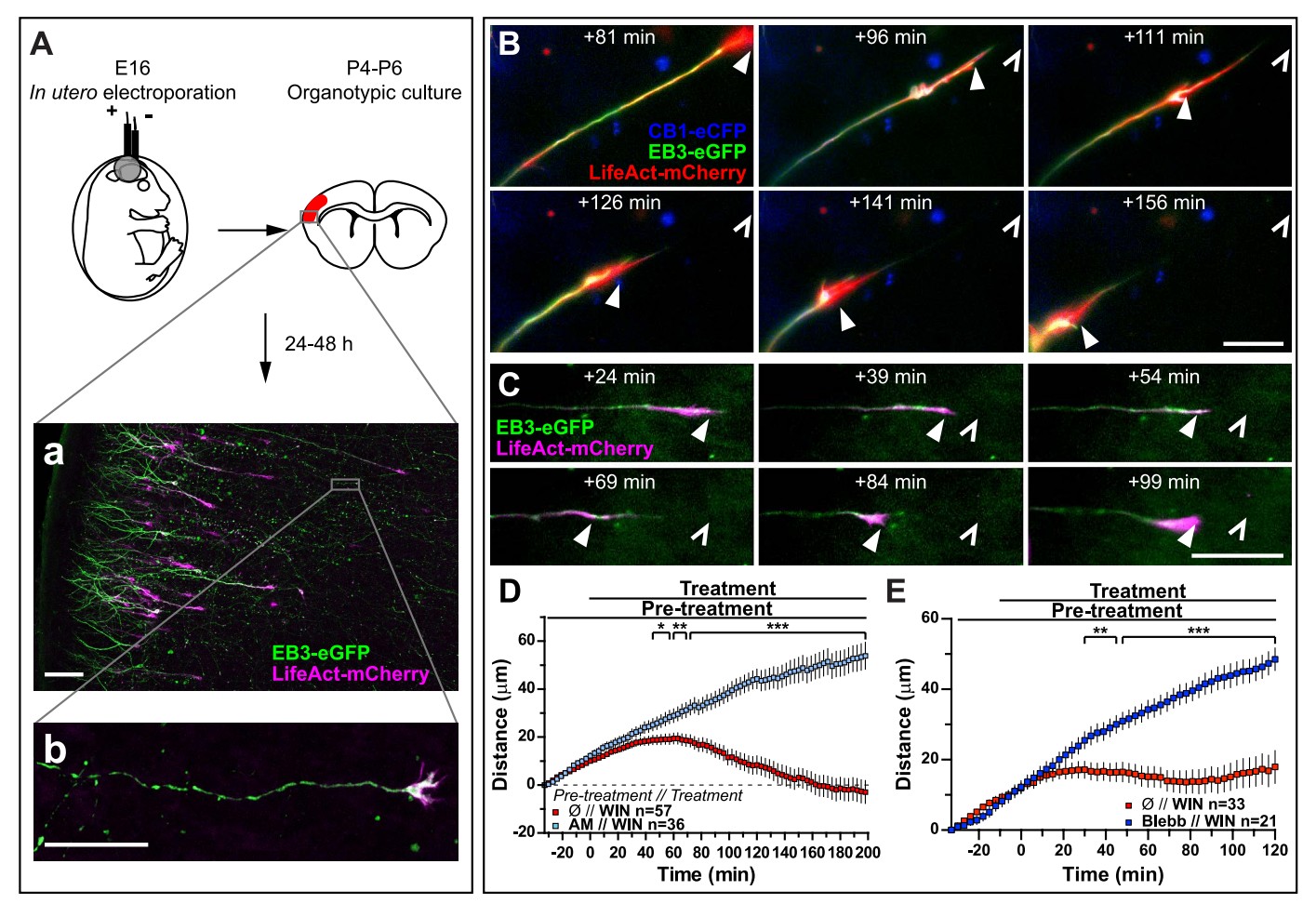

**Figure 4**. Activation of exogenous or endogenous CB1Rs modifies growth cone dynamics ex vivo. Progression of dynamic, F-actin-rich corticofugal growth cones from organotypic slices cultured for 24 to 48 hr, prepared from P4-6 rat brains, previously electroporated in utero at E16 to express EB3-eGFP, LifeAct-mCherry, with or without Flag-CB1R-eCFP, was followed by time-lapse imaging. (**A**) Experimental design and illustration of a typical transfected cortical area (**A**) and of a typical labeled growing axon (**B**). For the illustration, the organotypic section was fixed and EB3-eGFP signal was enhanced by incubation with an anti-GFP antibody. (**B–E**) Response to CB1R agonist WIN (1 μM, added on 0 min). The F-actin-rich growth cone is indicated by arrowheads. Open arrowheads indicate growth cone position at 0 min (**B**, **D**) WIN-induced retraction in growth cones expressing EB3-eGFP, LifeAct-mCherry, and Flag-CB1R-eCFP is abolished by pre-treatment with 5 μM CB1R-specific antagonist AM281. (**C**, **E**) WIN-induced retraction in growth cones expressing EB3-eGFP and LifeAct-mCherry is abolished by pre-treatment with blebbistatin (25 μM). Results are pooled from at least two independent experiments and are expressed as mean ± SEM. *p < 0.05; **p < 0.01; ***p < 0.001, calculated using Student's t-test. Scale bar: 100 μm on **A**, 20 μm elsewhere.

## Discussion

Our results show that acute CB1R activation results in rapid contraction of the neuronal actomyosin cytoskeleton. CB1R acts through heterotrimeric $G_{12}/G_{13}$ proteins, Rho GTPase, and ROCK to induce the contractile interaction of NM II with F-actin. This contraction triggers the retraction of the actin-rich growth cone of the most distal 60–70 μm of the axon in cultured hippocampal neurons and in cortical neurons in organotypic slices. Pharmacological inhibition of either CB1Rs or NM II during brain development leads to excessive growth of corticofugal axons in vivo, suggesting that CB1R-induced actomyosin contractility is necessary for the correct pathfinding by mediating their repulsion from the sub-ventricular zone. This contractile behavior is not limited to the growth cone since CB1R-induced actomyosin contraction leads to neurite retraction, cell rounding, and a significant elevation in cell rigidity in the Neuro2A cells. Similarly, in the somatodendritic region of cultured

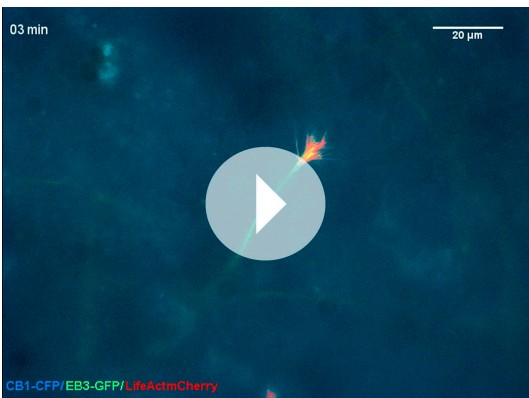

03 min                                    20 μm

CB1-CFP/EB3-GFP/LifeActmCherry

**Video 5**. CB1R activation induces retraction of actin-rich growth cones in organotypic slices. Dynamic, F-actin-rich corticofugal growth cones from organotypic slices were cultured for 24 to 48 hr, prepared from P4-6 rat brains, previously electroporated with EB3-eGFP, LifeAct-mCherry, and Flag-CB1R-eCFP in utero (See *Figure 3*). Treatment with 1 μM WIN at 30 min induces retraction of the growth cone. Scale bar: 20 μm.

hippocampal neurons, distal regions of dendrites retract while proximal parts broaden. Finally, ROCK and NM II mediate the inhibitory effect of chronic CB1R activation on dendrite development, by increasing the natural contractile tone of neurons.

Owing to its position downstream of convergent signaling pathways, the NM II protein plays a pivotal role in the control of tissue architecture through its participation in processes that require cell reshaping and movement, such as cell adhesion, cell migration, and cell division (*Vicente-Manzanares et al., 2009*). In neurons, NM II is also involved in diverse aspects of cell movement, such as neuronal migration and the structural organization and efficient extension of the growth cone, which requires an intricate balance between dynein, microtubules, actin, and different myosin II isoforms (*Vallee et al., 2009*). Remarkably, previous in vitro results have shown that NM II is important for turning in response to boundaries of substrate-bound laminin-1 (*Turney and Bridgman, 2005*) and that pharmacological inhibition or genetic silencing of NM II leads to disorganization of the growth cone, allowing rapid axon extension over inhibitory substrates (*Hur et al., 2011*). In the present study, we report a comparable in vivo effect, by showing that blebbistatin treatment leads to elevated axonal invasion of the embryonic sub-ventricular zone (SVZ). This territory, which is populated by proliferating neuronal progenitors, is typically avoided by corticofugal axons during their progression towards sub-cortical target zones. To our knowledge, these results show for the first time the existence of NM II-mediated axonal repulsion in vivo, suggesting that mobilization of NM II participates in the correct guidance of corticofugal axonal projections. Since pharmacological CB1R blockade has similar effects to NM II inhibition (i.e., excessive axonal growth), a likely scenario suggests that the endocannabinoid 2-AG, whose synthesizing enzyme DAGLα is specifically expressed at high levels by proliferating progenitor cells of the SVZ (*Goncalves et al., 2008*), acts through CB1Rs to repulse invading corticofugal axons through NM II-mediated growth cone retraction.

CB1Rs also rapidly modify the morphology of Neuro2A cells and cultured hippocampal neurons through enhanced actomyosin contractility, leading to large-scale reorganization of neuronal compartments that contain F-actin. Notably, CB1Rs not only alter the internal organization of the growth cone, as suggested previously (*Berghuis et al., 2007*; *Argaw et al., 2011*), but cause the retraction of the distal axon over several tens of microns, both in cultured hippocampal neurons and in cortical neurons in organotypic slices. This NM II-dependent contraction leads to the characteristic periodic bending of microtubules. This particular phenotype was similarly observed during strong NM II-mediated retraction in DRG neurons after the activation of the LPA receptor (*Bouquet et al., 2007*) or after treatment with Sema 3A (*Gallo et al., 2002*; *Wylie and Chantler, 2003*; *Gallo, 2006*;). In addition, nitric oxide, widely recognized to induce axonal retractions during development (*Cramer et al., 1998*; *Ernst et al., 2000*), was reported to induce similar rapid axonal retraction accompanied by periodic bends (*He et al., 2002*).

In addition, RhoA, ROCK, and NM II are known constitutive inhibitors of neurite development (*Kollins et al., 2009*). By activating the CB1R/$G_{12}$/$G_{13}$/Rho GTPase/ROCK/NM II axis characterized in our study, endo- or exogenous cannabinoids are likely to mobilize a widely employed myosin-activating machinery that is involved in growth cone navigation and in the establishment and maintenance of neuronal morphology. Interestingly, similar $G_{12}$/$G_{13}$-dependent signaling mechanism is mobilized downstream of at least two developmentally implied neuronal GPCRs, the LPA receptor (*Katoh et al., 1998*; *Kranenburg et al., 1999*) and GPR55, a putative 'atypical' cannabinoid receptor (*Ryberg et al., 2007*; *Sharir and Abood, 2010*), through activation by lysophosphatidylinositols but not through genuine cannabinoid ligands (*Obara et al., 2011*). Therefore, coupling of a bona fide neurotransmitter and drug receptor, such as CB1R, to this major developmental pathway may open interesting research and therapeutic perspectives.

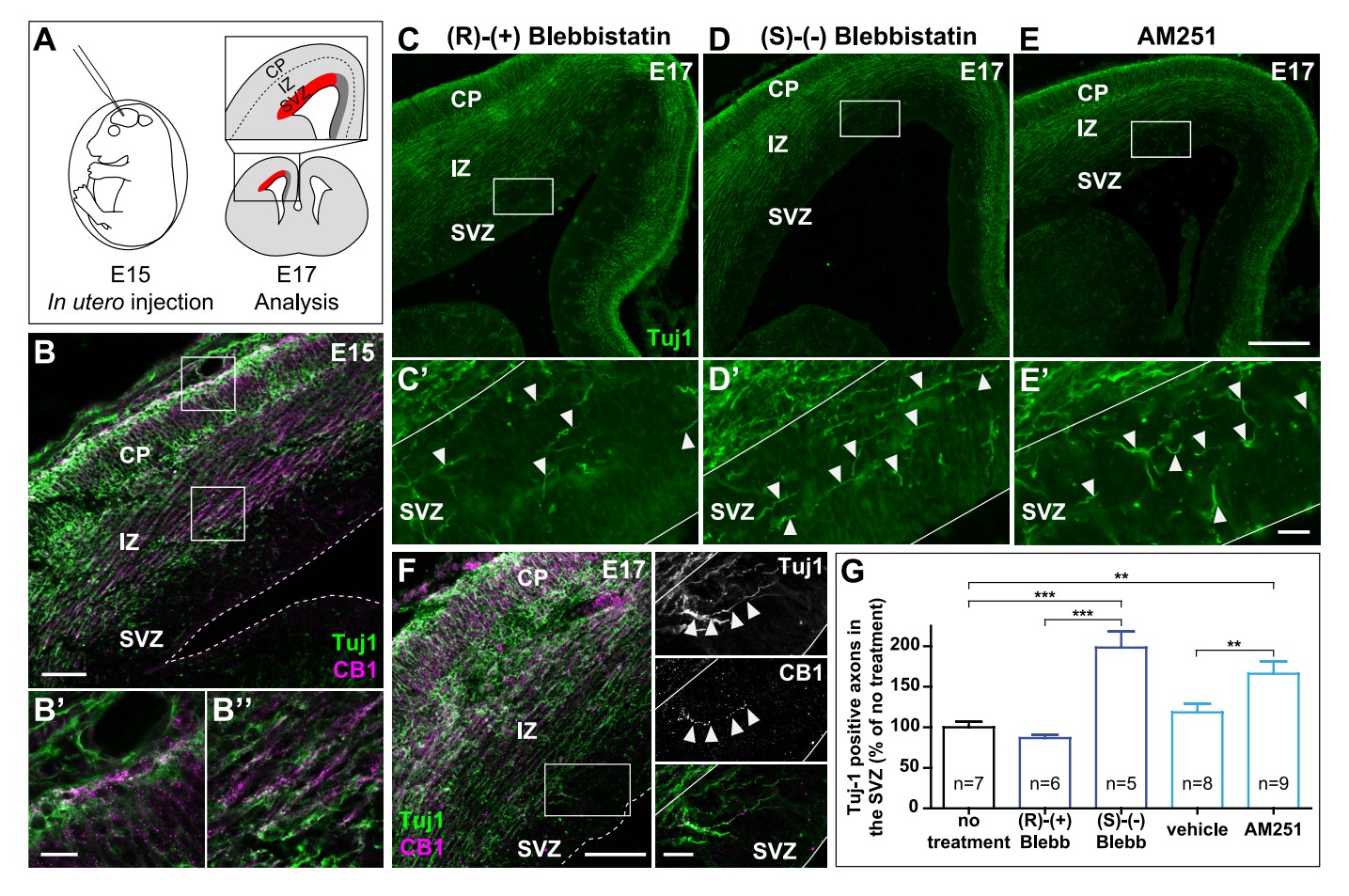

**Figure 5**. Actomyosin contractility is required for the correct targeting of CB1R expressing corticofugal axons. (**A**) Experimental design. Left: in utero intracerebroventricular injection of E15 rat embryos. Right: analysis of axons in the lateral sub-ventricular zone (SVZ, red). (**B**) E15 corticofugal axons starting from the cortical plate (CP) and progressing through the intermediate zone (IZ) highly co-express Tuj-1 (green) and CB1R (magenta) and mostly avoid the SVZ. (**C**–**G**) In embryos injected with 1 μl of the active NM II-ATPase inhibitor (S)-(−)-blebbistatin (250 μM) (**D**, **D′**), or with AM251 (1 mM) (**E**, **E′**), but not with the inactive (R)-(+) stereoisomer (250 μM) or the vehicle of AM251 (DMSO 2.8%) (**C**–**C′**), there is a significant increase of mistargeted corticofugal axons in the lateral SVZ (arrowheads, **G**). (**F**) Expression of endogenous CB1Rs in a representative Tuj1 positive axon invading the SVZ (arrowheads) from an embryo treated with active (S)-(−)-blebbistatin. Results are pooled from three independent experiments and are expressed as mean ± SEM, **p < 0.01, ***p < 0.001 calculated using one-way ANOVA followed by Newman–Keuls post-tests. Scale bars: 100 μm on **B** and **F** (left), 250 μm on **C**, **D**, and **E** and 25 μm on **B′**, **B″**, **C′**, **D′**, **E′**, and **F** (right).

The following figure supplement is available for figure 5:

**Figure supplement 1**. Cortifugal origin of Tuj1-expressing axons in the SVZ.

NM II is also involved in integrin-mediated cell adhesion; in turn, the adhesive properties of the substrate also control NM II activation (*Vicente-Manzanares et al., 2009*). However, CB1R-mediated morphological effects reported in the present study may not result from reduced neuronal adhesion, considering the time-scale of the rapid neuronal retraction. Instead, retracting the Neuro2A cells and growth cones of hippocampal neurons both in vitro and ex vivo leave behind thin membranous fibers, which contain F-Actin and are still attached to the adhesive substrate. The formation and morphology of these fibers are similar to those of retraction fibers reported in mitotic cells (*Cramer and Mitchison, 1995*) also generated by rapid contraction of the cellular actomyosin cortex (*Théry and Bornens, 2008*).

The ensemble of these results combined with the bulk of the available experimental data (reviewed in *Gaffuri et al., 2012*) suggests that endocannabinoids acting through CB1Rs exert a general negative effect on cell spreading and neurite growth. Basal cell-autonomous or paracrine activation of CB1Rs would yield relatively weak tonic inhibition of growth in the majority of developmental

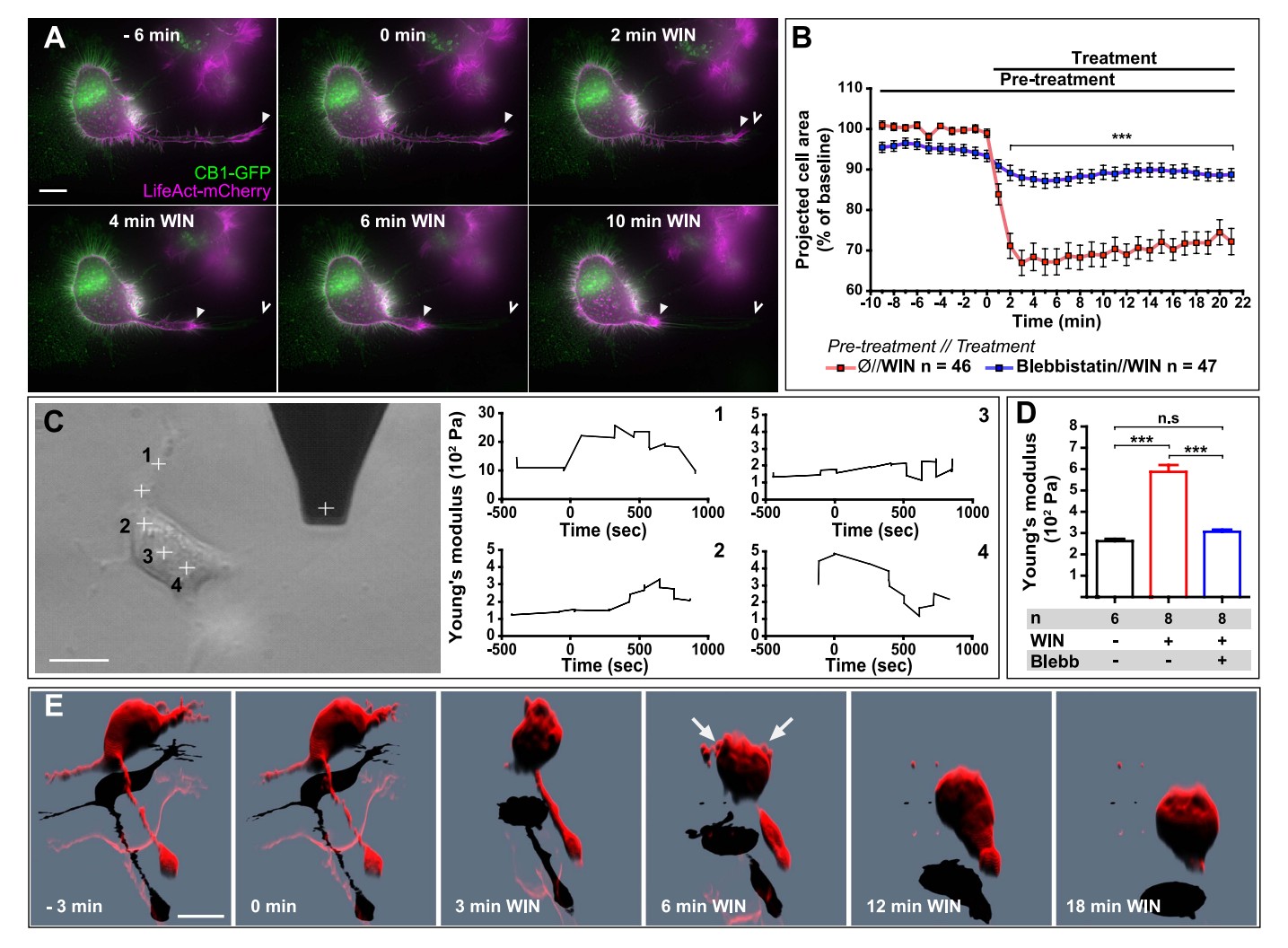

**Figure 6**. CB1R-induced actomyosin contraction results in neurite retraction and transiently increased cell stiffness in Neuro2A cells. (**A** and **B**) Cells expressing Flag-CB1R-eGFP and LifeAct-mCherry. F-actin accumulates in the extremity and shaft of neurites (arrowheads). Agonist WIN (100 nM) induces retraction of neurites. Open arrowheads: neurite tip at 0 min. (**B**) Blebbistatin (25 μM) significantly reduces 100 nM WIN-induced cell rounding. Results are expressed as mean ± SEM. (**C**) Phase-contrast image of a Neuro2A cell and the AFM cantilever. Stiffness response to 100 nM WIN at different cell locations (crosses). Subsequent measurements were focused on the cell bodies, corresponding to positions 2 and 3. (**D**) Blebbistatin (25 μM) significantly reduces 100 nM WIN-induced increase of cell stiffness. Results are pooled from at least three independent experiments and are expressed as mean stiffness between 2 and 8 min after stimulation ±SEM. (**E**) 3D reconstruction shows neurite retraction, cell rounding, and transitory blebbing (arrows) following WIN treatment (100 nM). n.s p > 0.05; ***p < 0.001, calculated using Student's t-test on B and using one-way ANOVA followed by Newman–Keuls post-tests on (**D**). Scale bars: 10 μm on (**A**) and (**E**), 15 μm on (**C**).

settings in all neuronal sub-regions where F-actin is present. Local growth-promoting effectors at the growth cone, such as the self-amplifying autocrine promoter BDNF (*Cheng et al., 2011*) or netrin-1 (*Argaw et al., 2011*), may locally surmount this weak negative tone. The resulting 'channeling' effect of widespread CB1R-mediated weak inhibition would help the neuron to focus its resources to a limited amount of growth locations, leading to more efficient polarized growth. Such weak inhibition may serve also to coordinated guidance of axonal fascicles in the brain where moderate production of eCBs by nearby axons would be used as a repulsion cue that helps axons to grow straightly towards their target. However, when growth cones reach a region highly enriched with eCBs, such as the sub-ventricular zone, enhanced eCB signaling could result in growth cone arrest, repulsion, or collapse, efficiently steering out CB1R-expressing axons from these areas. Finally, CB1R-induced actomyosin contractility may also contribute to establish functionally adequate somatodendritic

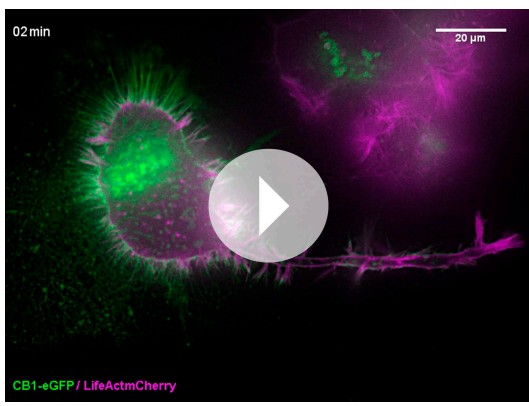

**Video 6**. CB1R activation induces neurite retraction and cell rounding in Neuro 2A cells. Neuro2A cell expressing Flag-CB1R-eGFP and LifeAct-mCherry. Treatment with 100 nM WIN at 30 min induces neurite retraction and cell rounding. Scale bar: 20 μm.

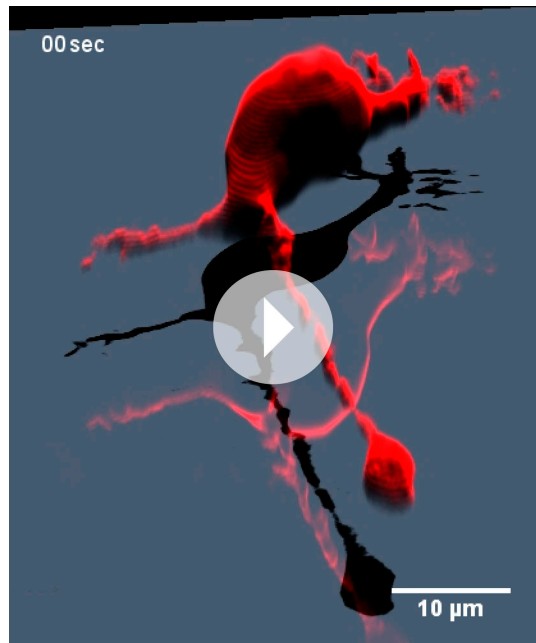

**Video 7**. CB1R activation induces neurite retraction, cell rounding, and temporary blebbing in Neuro 2A cells. 3D reconstruction of a Neuro2A cell expressing Flag-CB1R-eGFP and DsRed2. Treatment with 100 nM WIN at 7 min (420 s) induces neurite retraction, cell rounding, and transitory blebbing. Scale bar: 10 μm

morphology, acting as a negative regulator of dendritic growth.

In our study, we were able to characterize in detail CB1R-induced actomyosin contraction, which is rather subtle and transitory downstream of endogenous GPCRs, by using high-resolution time-lapse imaging, atomic force microscopy, and moderate over-expression of CB1Rs. Consequently, the amplitudes of reported cytoskeleton changes are likely dramatic compared to CB1R-induced remodeling in typical physiological settings. Nevertheless, the results concerning endogenous CB1Rs, obtained in cultured neurons, in organotypic slices, and in vivo suggest physiological relevance for our findings.

In conclusion, we identify NM II-mediated actomyosin contraction as a mechanism conveying a wide-ranging inhibitory role for cannabinoids in neuronal expansion and growth, downstream of CB1R coupled to $G_{12}/G_{13}$ proteins and the Rho-associated kinase ROCK. Such modulation of the neural actomyosin cytoskeleton has not yet been reported downstream of neurotransmitter GPCRs, therefore our results open previously unexpected perspectives in the study and comprehension of brain function.

## Materials and methods

### Chemicals and antibodies

CB1R agonists WIN55,212-2, CP-55940, HU-210, 2-arachidonoylglycerol (2-AG), and CB1R-specific antagonist/inverse agonists AM281 and AM251 were acquired from Tocris Bioscience (Bristol, UK). Rho-associated kinase inhibitor Y-27632 and nocodazole were purchased from Calbiochem (San Diego, CA). Blebbistatin, cytochalasin D, ML-7, Δ⁹-Tetrahydrocannabidiol solution (THC), and pertussis toxin (PTX) were brought from Sigma (Saint-Louis, MO). The Rho-GTPase inhibitor C3 transferase (C3T) was purchased from Cytoskeleton, Inc. Mouse anti-neuron-specific beta III tubulin (Tuj-1) antibody was obtained from Sigma (Catalog Number T8660), rabbit anti-myosin phospho S19/phospho S20 antibody was obtained from Rockland (Gilbertsville, PA, Cat. no. 600-401-416), rabbit anti-CB1R antibody was produced by Eurogentec (Seraing, Belgium) and described previously (*Thibault et al., 2013*), and chicken anti-GFP antibody was from AVES (Tigard, OR). Alexa-Fluor-conjugated secondary antibodies were purchased from Life Technologies (Carlsbad, CA). All culture media and additives were from PAA Laboratories (Pasching, Austria).

### DNA constructs

The DsRed2 encoding plasmid was produced by Clontech (Mountain View, CA). The CB1R-eCFP (Enhanced Cyan Fluorescent Protein) and Flag-CB1R-eGFP constructs have previously been described

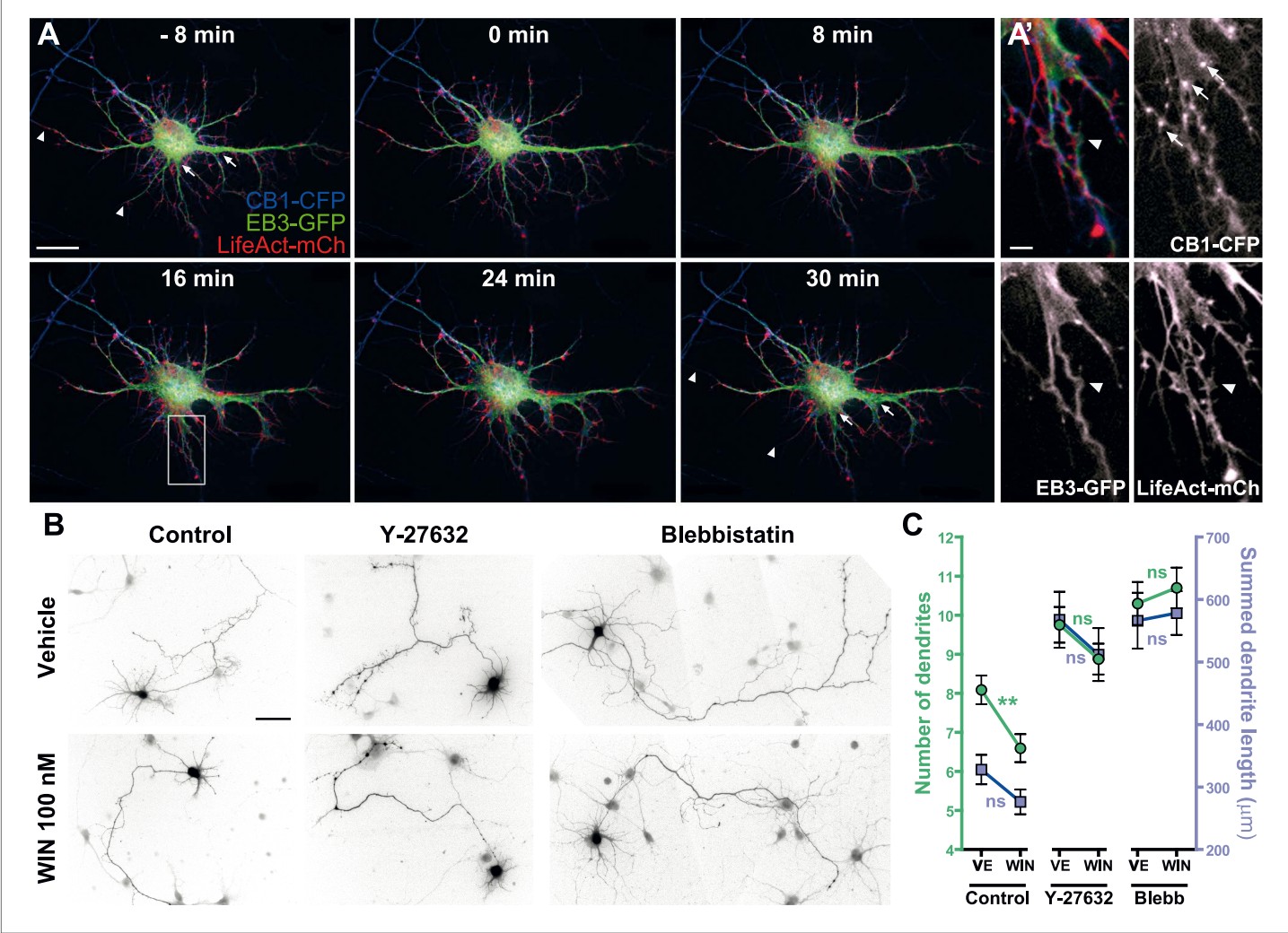

**Figure 7**. Acute and chronic effects of CB1R-mediated actomyosin contraction on somatodendritic morphology. (**A**) Cultured hippocampal neurons expressing CB1R-eCFP, LifeAct-mCherry, and EB3-eGFP at DIV8. Application of 100 nM WIN results in rapid and significant reorganization of somatodendritic morphology, characterized by retraction of distal dendritic parts (arrowheads), and broadening of the proximal part of dendrites (arrows). (**A'**) In dendrites, characteristic microtubule bending (arrowheads) and appearance of straight cable-like F-actin bundles (arrowheads) are accompanied by CB1R endocytosis after agonist activation (arrows). (**B** and **C**) Chronic inhibition of ROCK or NM II abolishes CB1R-activation induced changes structure of the cultured hippocampal neurons expressing Flag-CB1R-eGFP and the structural marker DsRed2 at DIV4. Cells were fixed at 24 hr after treatment with inhibitors of ROCK (Y-27632, 10 μM) or NM II (blebbistatin, 25 μM) in the presence of vehicle (VE) or CB1R agonist WIN (100 nM). A representative cell is shown for each condition. (**C**) Results are pooled from at least two independent experiments and are expressed as mean ± SEM. n.s p > 0.05; **p < 0.01, calculated using one-way ANOVA followed by Newman–Keuls post-tests. Scale bars: 20 μm on (**A**), 5 μm on (**A'**), and 50 μm on (**B**).

elsewhere (*Leterrier et al., 2004*, *2006*). LifeAct-mCherry was a kind gift of G Montagnac and P Chavrier (Institut Curie, Paris, France). pEGFP-N3–EB3 plasmid was a kind gift of M Piel (Institut Curie, Paris, France). pCAG-Cre and pCALNL-GFP in which GFP was replaced by Flag-CB1R-eCFP, LifeAct-mCherry, or eGFP-EB3 sequences for in utero electroporation experiments were a kind gift from T Matsuda and C Cepko (Harvard Medical School). All constructs were verified by full-length sequencing.

## RNA interference

For silencing rat *non-muscle myosin IIA*, rat *non-muscle myosin IIB*, rat $G_{12}$- and rat $G_{13}$- specific SMARTpools were chemically synthesized by Dharmacon Research (Lafayette, CO) and siRNA targeting

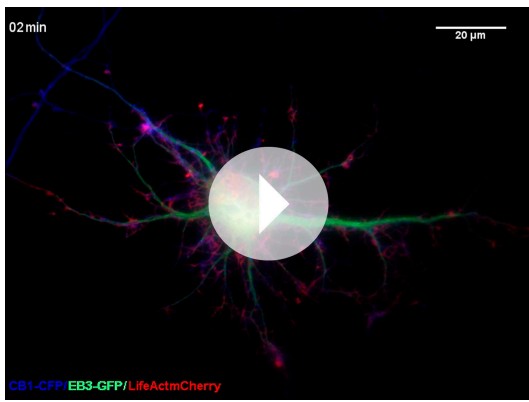

02 min                                                          20 μm

CB1-CFP/EB3-GFP/LifeActmCherry

**Video 8**. CB1R activation induces rapid remodeling of the somatodendritic region in cultured hippocampal neurons. Somatodendritic region of a cultured hippocampal neuron co-expressing CB1R-eCFP, LifeAct-mCherry, and EB3-eGFP at DIV8. The axon, whose initial segment is typically strongly labeled with EB3-GFP, exits the frame in the upper-left corner. The F-actin-rich growth cone, such as shown in **Video 1**, is at the growing end of the axon, typically hundreds of microns away from the soma at DIV8. Treatment with 100 WIN at 10 min induces retraction of distal dendrites and broadening of proximal dendrites. Scale bar: 20 μm. DOI: 10.7554/eLife.03159.024

luciferase (CGUACGCGGAAUACUUCGA, Proligo-Sigma) was used as a control, as described previously (*Miserey-Lenkei et al., 2010*).

## Cell cultures

Neuro2A cells (ATCC CCL-131) were grown in DMEM (Life Technologies) supplemented with 4.5 g/l glucose, GlutaMAX I (Life Technologies), 10% fetal bovine serum, 10 U/ml penicillin G and 10 mg/ml streptomycin. Neuronal cultures were prepared as described previously (*Carrel et al., 2011*). Briefly, hippocampi of rat embryos were dissected at embryonic days 17–18. After trypsinization, tissue dissociation was achieved with a Pasteur pipette. Cells were plated on poly-D-lysine-coated coverslips at a density of 60,000–75,000 cells per 15 mm coverslip and cultivated in complete Neurobasal (Life Technologies) medium supplemented with B27 (Life Technologies), containing 0.5 mM L-glutamine, 10 U/ml penicillin G, and 10 mg/ml streptomycin containing conditioned medium obtained by incubating glial cultures (70–80% confluency) for 24 hr. Experiments were performed in agreement with the institutional guidelines for the use and care of animals and in compliance with national and international laws and policies (Council directives no. 87-848, 19 October 1987, Ministère de l'Agriculture et de la Forêt, Service Vétérinaire de la Santé et de la Protection Animale).

## Cell transfections

Neuro2A cells were transfected, in 6-well plates for ATF and deconvolution or in 12-well plates for video microscopy, with 0.8 μg of plasmid DNA using Effectene reagent (Qiagen, Venlo, NL) and processed 24 hr after transfection.

Hippocampal neurons were transfected on DIV3 (for morphometry) or DIV5-8 (for videomicroscopy) as follows: for each coverslip, plasmid DNA (2 μg) and Lipofectamine 2000 (1.25 μl, Life Technologies) in Neurobasal medium were combined and incubated for 30 min. After the addition of complete Neurobasal medium containing B27 supplement, the mix was applied onto the neuronal culture for 3 hr at 37°C. Receptor expression was allowed in growth medium for 24 to 72 hr after transfection. Immediately after transfection, DIV3 transfected hippocampal neurons were incubated with different pharmacological treatments and fixed after 24 hr. Our transfection protocol leads to moderate over-expression of CB1Rs and we imaged only low-expressing neurons in which sub-neuronal traffic and targeting of transfected receptors is similar to that of endogenous CB1Rs (*Leterrier et al., 2006*; *Vitalis et al., 2008*; *Thibault et al., 2013*).

For siRNA transfections, two different mixes were prepared: one with Lipofectamine (2 μl) and plasmid DNA (1.25 μg of Flag-CB1-eGFP and 1.25 μg of LifeAct-mCherry) in 50 μl of Neurobasal medium and one with Lipofectamine and siRNA (2.4 μl of each siRNA at 50 μM alone or combined with other siRNAs were mixed in 50 μl of Neurobasal medium). In controls, appropriate volumes of anti-luciferase siRNA (50 μM) were used to match the total amount of transfected siRNAs. After 30 min of incubation, the two mixes were combined, completed to 250 μl with conditioned complete Neurobasal medium containing B27 supplement and applied to the neuronal culture for 3 hr at 37°C. At the end of incubation, the mix was replaced by fresh complete Neurobasal medium and neurons were used 48 to 72 hr later.

## Animals

Animals were housed individually with free access to food and water and maintained in a temperature-controlled environment on a 12 hr light/dark cycle. Experiments were performed in agreement with

the institutional guidelines for the use and care of animals and in compliance with national and international laws and policies (Council directives no. 87-848, 19 October 1987, Ministère de l'Agriculture et de la Forêt, Service Vétérinaire de la Santé et de la Protection Animale).

## In utero electroporation, preparation of organotypic slices, and histological procedures

Pregnant Sprague–Dawley rats at gestation day 16 were anesthetized with Ketamine/Xylazine (75/10 mixture). The abdominal cavity was opened to expose the uterine horns. 1–3 µl of plasmids (0.5 µg/µl for pCAG-Cre, 1 µg/µl for pCALNL-LifeAct-mCherry and pCALNL-EB3-eGFP, and 1.5 µg/µl for pCALNL-Flag-CB1R-CFP) with 1 mg/ml Fast Green (Sigma) were microinjected through the uterus into the lateral ventricles of embryos by pulled glass capillaries (Drummond Scientific, Broomall, PA). Electroporation was performed by placing the heads of the embryos between tweezer-type electrodes. Square electric pulses (65 V, 50 ms) were passed five times at 1 s intervals using a CUY21 EDIT electroporator (Nepa Gene, Chiba, Japan).

For axonal localization analysis, rat brains (E20) were dissected and fixed for 48 hr in 4% paraformaldehyde (PFA) in PBS at 4°C. Brains were then cryoprotected in 30% sucrose in PBS, frozen in OCT compound (Sakura, Tokyo, Japan), and sectioned coronally at 16 µm using a cryostat.

For organotypic slice preparation, rats were sacrificed at P4-6 by decapitation under deep anesthesia with pentobarbital. Brains were dissected and transferred into liquid 3% low-melting agarose (38°C) and placed on ice. Embedded brains were cut coronally (300 µm) with a VT1000S vibratome (Leica, Nussloch, Germany) at 4°C. Slices were transferred onto sterilized culture plate inserts (0.4-µm pore size, Millicell-CM, Millipore, Billerica, MA) and cultured in semidry conditions in a humidified incubator at 37°C under 5% $CO_2$ atmosphere in wells containing Neurobasal medium (Life Technologies) supplemented with 1% B27 (vol/vol), 1% N2 (vol/vol), 1% GlutaMAX I (vol/vol), and 1% penicillin/streptomycin (vol/vol, Life Technologies). Slices were cultured for 24–48 hr before videomicroscopy. For illustration of electroporated cortical area, some organotypic slices cultured for 24 hr were fixed for 2 hr in 4% PFA in PBS.

## In utero cerebroventricular injections and histological procedures

Pregnant Sprague–Dawley rats at gestation day 15 were prepared as for in utero electroporation. Then, 1 µl of a solution containing active NM II ATPase inhibitor (S)-(−)-blebbistatin (250 µM), inactive (R)-(+) stereoisomer (250 µM), AM251 (1 mM), or 2.8% DMSO (vehicle for AM251), mixed with 1 mg/ml Fast Green were microinjected through the uterus into the lateral ventricles of embryos by pulled glass capillaries. Embryos were allowed to develop in utero for 2 days. E17 brains were then dissected, fixed for 48 hr in 4% paraformaldehyde (PFA) in PBS at 4°C, cryoprotected in 30% sucrose in PBS, frozen in OCT compound, and sectioned coronally at 20 µm using a cryostat.

## Immunofluorescence

For immunohistochemical staining of brain sections or fixed organotypic brain slices, sections were incubated with a combination of mouse anti-Tuj1 antibody, C-Ter rabbit antibody, and chicken anti-GFP antibody (each diluted at 1:1000) overnight at room temperature in PBS (0.02 M) containing 0.3% Triton and 0.02% sodium azide (PBS-T-azide). For immunofluorescence detection of phosphoMLC, cultured neurons were fixed for 15 min in 4% PFA with 4% sucrose, permeabilized with PBS-T-azide and incubated for 90 min with the anti-phosphoMLC antibody (1:1000) diluted in 2% Bovin Serum Albumin and 3% Normal Goat Serum.

Following washes, sections or coverslips were incubated with the appropriate secondary antibodies for 2 hr at room temperature and coverslipped with Mowiol mounting medium.

## Microscopy

For time-lapse microscopy, coverslips were placed in a Ludin chamber (Life Imaging Services, Basel, Switzerland) filled with imaging buffer (120 mM NaCl, 3 mM KCl, 2 mM $CaCl_2$, 2 mM $MgCl_2$, 10 mM glucose, 10 mM HEPES, and 2% B27, pH 7.35, 250 mOsm to match culture growth medium) (*Lu et al., 2007*). Wide-field images were taken on a motorized Nikon Eclipse Ti-E/B inverted microscope with the Perfect Focus System (PFS) in a 37°C chamber, using an oil immersion CFI Plan APO VC 60x, NA 1.4 objective (Nikon, Melville, NY), equipped with a Polychrome V monochromator (Till Photonics, Gräfelfing, Germany) and an Intensilight light source (Nikon), a CoolSnap HQ2 camera (Photometrics, Tucson, AZ), and piloted by Metamorph 7.7 (Molecular Devices, Sunnyvale, CA). All filter sets were

purchased from Semrock (Rochester, NY) and the absence of cross-talk between different channels was checked with selectively labeled preparations.

For the evaluation of neurite retraction in vitro, neurons or Neuro2A cells co-expressing CB1R-eCFP/EB3-eGFP/LifeAct-mCherry or Flag-CB1R-eGFP/LifeAct-mCherry or expressing LifeAct-mCherry alone were imaged every 2 min in each corresponding detection channel, and the mCherry detection channel was used for quantification. Treatments with inhibitors were applied on transfected cells 20 min before stimulation with the agonist. Blebbistatin treated cells were only illuminated through the mCherry excitation channel, in order to avoid phototoxic effects of lower illumination wavelengths. For the evaluation of axon retraction ex vivo, neurons co-expressing Flag-CB1-CFP/EB3-eGFP/LifeAct-mCherry or expressing LifeAct-mCherry were imaged every 3 min for 150–240 min, and inhibitor treatments were applied 30 min before agonist stimulation. For pharmacological treatments, ligands dissolved in dimethylsulfoxide were added directly to the culture medium. The highest final concentration reached was 0.2% DMSO; control experiments with up to 0.5% DMSO have shown the absence of effects on neuronal morphology and on the cellular distribution of CB1Rs.

For the analysis of cortifugal axon development, images of labeled rat brain sections were taken on a Zeiss AxioImager M1 microscope using a 40× 0.75 numerical aperture (NA) objective. In each experiment, all acquisitions were performed using strictly identical exposure conditions. For the analysis of the images the SVZ was delimited and the corticofugal axons present in it were counted in blind. Between 5 and 9 embryos were employed per condition analyzing a mean of 9 brain slices per animal.

For morphological analysis, widefield images were taken on a Zeiss Imager M1 microscope with dry 20× NA 0.75 and 40× NA 0.75 and oil-immersion 100× NA 1.3 objectives (Zeiss, Oberkochen, Germany). In all cases, emission and excitation filters proper to each fluorophore were used sequentially and the absence of cross-talk between different channels was checked with selectively labeled preparations. Neurites were outlined and measured using an assisted semiautomatic method (NeuronJ) (*Meijering et al., 2004*). For neurons at DIV4, primary and secondary dendrites were outlined and their number and length were measured. Retraction of neurites was determined using Metamorph.

For the CB1/Tuj1 and Tuj1/GFP co-localization experiments, and for illustration of the electroporated cortical area, images were taken on a Nikon A1 laser-scanning confocal microscope with dry 10× NA 0.30 and 20× NA 0.75 and oil-immersion 60×, NA 1.4 objectives.

## Atomic force microscopy measurement and processing

Novascan (Ames, IA) cantilevers with attached $SiO_2$ spherical beads (1-μm diameter) and nominal spring constant 0.06 N/m were used. Photodiode sensitivities of each cantilever were calibrated before and after measurements on the stiff surface region of culture dishes. The cantilever spring constant was determined using the thermal fluctuations method implemented in the Nanoscope 8 software (*Hutter, 1993*).

Measurements were carried out 1 day after seeding Neuro2A cells at 37°C on a commercial AFM (Catalyst, Bruker, Billerica, MA) mounted on an inverted optical microscope (Olympus, Tokyo, Japan). We obtained force–distance (F–z) curves of ~3 μm peak-to-peak amplitude at 0.5 Hz, ~3 μm/s. The maximum relative deflection (d) was controlled to reach an indentation depth of <400 nm. We placed the cantilever tip around the center of the cells with the help of optical images of the tip and samples acquired with a CCD camera (Hamamatsu, Shizuoka Pref., Japan).

Three different cells were probed in a single AFM session by acquiring force curves at time intervals of <1 min. The same spherical tip was used in all measurements. Measurements were carried out before and after addition of vehicle or 100 nM of WIN at time point 0. Blebbistatin was applied 20 min before time point 0.

Each experiment approaching F–z curve was fitted by the Hertz model of a sphere indenting an elastic half space (*Rico et al., 2005*): $F = \frac{4}{3}\frac{E}{1-v^2}\sqrt{R\delta^{3/2}}$, where E being the Young's modulus, $v$ the Poisson ratio (0.5), R, the radius of the sphere, and, δ the indentation, which was calculated in terms of the point of contact ($z_c$) and deflection offset ($d_0$) as $\delta = z - z_c - (d - d_0)$.

## Deconvolution and surface reconstruction

To follow shape change of retracting Neuro2A cells co-expressing Flag-CB1R-eGFP and the structural marker DsRed2, high-resolution images were acquired as a three-dimensional time series. For 61 time

frames separated by 30 s, 51 z-slices of dimensions 149.64 µm × 111.8 µm (1392 pixels × 1040 pixels) and separated by 0.5 µm in height were captured. The signal to noise ratio of the images was improved by deconvoluting the z-stacks at each time frame by iteratively computing the maximum-likelihood deconvolved image using the Richardson–Lucy algorithm (Huygens Professional, Huygens, Inc., Hilversum, Netherlands). The stopping criteria for the algorithm was determined using a conservative estimate of the image quality improvement at each iteration, and approximately 60 iterations of the algorithm were required in order to significantly improve the image quality without introducing arte-facts into the deconvolved image. The surface of the deconvolved image stacks was computed at each time frame using a surface-rendering algorithm (FreeSFP, Huygens, Inc.).

## Statistical analysis

Data were analyzed using Prism (GraphPad Software, La Jolla, CA). Kolmogorov–Smirrow and Shapiro–Wilk tests were used to verify the normal distribution of the data. If the hypothesis of normality was confirmed, the significance of differences in mean was calculated using Student's *t*-test or one-way ANOVA followed by Newman–Keuls post-tests for p-value adjustment, elsewhere Kruskal–Wallis one-way ANOVA followed by Dunn's post-tests was used. For significance symbols, 'ns' means $p \geq 0.05$, one symbol means $p \leq 0.05$, two symbols mean $p \leq 0.01$, and three symbols mean $p \leq 0.001$. Outliers were removed when appropriate by applying Grubbs's test (ESD method [extreme studentized deviate]) available at the GraphPad QuickCalcs website: http://www.graphpad.com/quickcalcs/ConfInterval1.cfm (April 2014) or at NIST/SEMATECH e-Handbook of Statistical Methods, http://www.itl.nist.gov/div898/hand-book/eda/section3/eda35h1.htm, April 2014.

## Acknowledgements

We thank Mathieu Piel, Ana-Maria Lennon-Duménil, and Stéphanie Miserey-Lenkei (Institut Curie, Paris France), Christophe Leterrier (Laboratoire de Neurobiologie des Canaux Ioniques, INSERM UMR641, Marseille, France), and Arpiar Saunders (Harvard Medical School, Boston) for their insightful discussions, important technical advice, and comments on the manuscript. We thank Natalia Velez-Alicea (MIT, Boston) for her comments on the English syntax. ABR was supported by a doctoral fellowship from Fondation pour la Recherche Médicale, AR was supported by a postdoctoral fellowship from the Basque Country Government, DC was supported by a postdoctoral fellowship from the AXA Research Fund, AS was supported by a fellowship from the Fondation ARC. This work was supported by a grant to ZL from the French Agence Nationale de la Recherche (ANR-09-MNPS-004-01). The research in Dr SS' lab is supported by the Institut National de la Santé et Recherche Médicale (INSERM), the Aix-Marseille Université, and the French Agence Nationale de la Recherche (ANR).

## Additional information

### Funding

| Funder | Grant reference number | Author |
|---|---|---|
| Agence Nationale de la Recherche | ANR-09-MNPS-004-01 | Zsolt Lenkei |
| Institut national de la santé et de la recherche médicale | | Simon Scheuring, Zsolt Lenkei |

The funders had no role in study design, data collection and interpretation, or the decision to submit the work for publication.

### Author contributions

ABR, AR, DC, Conception and design, Acquisition of data, Analysis and interpretation of data, Drafting or revising the article; BMJ, FR, SS, Conception and design, Acquisition of data, Analysis and interpretation of data; AS, MH-C, JF, MHMF, Acquisition of data, Analysis and interpretation of data; ZL, Conception and design, Analysis and interpretation of data, Drafting or revising the article

### Ethics

Animal experimentation: Experiments were performed in agreement with the institutional guidelines for the use and care of animals and in compliance with national and international laws and policies

(Council directives no. 87-848, 19 October 1987, Ministère de l'Agriculture et de la Forêt, Service Vétérinaire de la Santé et de la Protection Animale). All surgery was performed under Ketamine/Xylazine anesthesia, and every effort was made to minimize suffering.

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
