## [Decision Letter]

Thank you for sending your work entitled “Cannabinoid-induced actomyosin contractility shapes neuronal morphology and growth” for consideration at *eLife.* Your article has been favorably evaluated by a Senior editor, a Reviewing editor, and 2 reviewers.

The Reviewing editor and the reviewers discussed their comments before we reached this decision, and the Reviewing editor has assembled the following comments to help you prepare a revised submission.

The manuscript by Roland and colleagues describes a series of studies examining the linkage between neuronal CB1 cannabinoid receptors (CB1Rs) and the myosin ii motor protein during CB1R agonist induced growth cone or dendrite retraction. The authors use a series of pharmacological or gene-silencing manipulations in vitro, ex vivo and in vivo to show that activation of CB1Rs leads to leads to changes in the neuronal cytoskeleton in retraction events that require myosin ii, rock and Galpha12/13. They nicely show through in vivo experiments that manipulation of CB1Rs and myosin ii retraction pathway they characterized in vitro may be required for extension of growth cones to their appropriate location within the developing cortex. Moreover, they elegantly use AFM to corroborate that changes in their pathway lead to predictable alterations in tension of the actin rich cortex in the Neuro2A cell system.

However, some aspects of the study are limiting the overall significance of the results. Endocannabinoids and their receptors (e.g. CB1R) have been already demonstrated to be involved in growth cone repulsion and directional guidance. The authors investigate further the mechanistic cytoskeletal process involving CB1R activation and growth cone retraction. By combining over-expression of CB1R in dissociated hippocampal neurons and pharmacological experiments, the authors claim that CB1R-actvivation results in rapid retraction of actin-rich growth cone domains through MLCK/ROCK-mediated NM II activation. Overall, the impact of the results would be significantly improved if they addressed the following points:

1) While the myosin II and rock inhibitor experiments are quite convincing functionally, a full physiological picture of how CB1Rs regulate myosin ii motor activity would require a careful examination of myosin II light chain phosphorylation. Do CB1Rs activate myosin light chain phosphorylation, as one would predict from the involvement of rock or Galpha12/13? Moreover, would Rho GTPase inhibitors also have the expected affect of retraction?

2) One of the major concerns is that the authors test the role of CB1R in growth cone motility by performing pharmacological experiments in hippocampal or cortical neurons by overexpressing CBR1. The authors claim that the results found in these overexpressing neurons are comparable to the effects found when activating endogenous (non-transfected) CBR1 neurons (Figure 1 and Figure 3). But this is by far not the case and thus is clearly questioning whether what they see is a relevant phenomenon that occurs under normal conditions. The retraction induced by WIN in non-CBR1 overexpressing neurons in vitro and in vivo its minimal and did not last in time in the presence of the agonist. Then, the rest of the experiments are describing what occurs in these CBR1-overexpressing neurons in vitro when cytoskeletal elements are being blocked. But, to this reviewer, it is questionable how relevant are the results found in these experiments to the endogenous role of CBR1 in vivo. Along the same line, the experiments of in utero electroporating CB1 in cortical neurons are not convincing as the WIN-induced retraction in growth cones expressing only EB3 and LifeAct is very mild and not comparable with the one found in CB1R axons, that incidentally becomes only evident 96 minutes after WIN addition.

3) Another major concern is the experiments and conclusions made by the authors when blocking NMII or CB1R in vivo by infusion of Blebbistatin or AM251 in the brain of rat embryos (Figure 4). The authors state “CB1R-expressing corticofugal axons showed important targeting errors, by invading the subventricular zone, from which CB1R-expressing axons are usually excluded”. Where do they see these errors? First, it is normal to find Tuj1 cut axonal elements in the SVZ in wild type conditions. Second, how do they know that these Tuj1 axons are corticofugal? How do they know that these Tuj1 axons express CB1R? From the data provided in this figure there is no evidence of targeting errors of corticofugal axons by blocking CBR1.

4) The authors use Neuro2A cells to investigate the role of CBR1 in other neuronal sub-compartments such as the cell soma. To this reviewer, this is not a clear step forward. Why do the authors choose to perform these experiments in a cell line instead of using the hippocampal neurons that they have been using so far? And more importantly, what is the relevance of their findings in this regard to the role of CB1R in neuronal development in vivo?

---

## [Author Response]

*1) While the myosin II and rock inhibitor experiments are quite convincing functionally, a full physiological picture of how CB1Rs regulate myosin ii motor activity would require a careful examination of myosin II light chain phosphorylation*. *Do CB1Rs activate myosin light chain phosphorylation, as one would predict from the involvement of rock or Galpha12/13? Moreover, would Rho GTPase inhibitors also have the expected affect of retraction?*

Thank you for this suggestion, which helped us to add interesting new data to our report. We have completed now a new series of experiments to directly show the rapid phosphorylation of myosin II light chain in the distal growth cone, both in neurons overexpressing CB1Rs or expressing endogenous CB1Rs. Strikingly, we detect rapid and strong upregulation of myosin light chain phosphorylation just behind the F-actin rich growth cone, at the right place for the subsequent rapid NMII-dependent contraction. We now show also that the Rho GTPase inhibitor C3 transferase has the expected inhibitory effect on WIN-induced retraction. These new results are now included in the new Figure 3.

*2) One of the major concerns is that the authors test the role of CB1R in growth cone motility by performing pharmacological experiments in hippocampal or cortical neurons by overexpressing CBR1. The authors claim that the results found in these overexpressing neurons are comparable to the effects found when activating endogenous (non-transfected) CBR1 neurons (*Figure 1
*and*
Figure 3*). But this is by far not the case and thus is clearly questioning whether what they see is a relevant phenomenon that occurs under normal conditions. The retraction induced by WIN in non-CBR1 overexpressing neurons in vitro and in vivo its minimal and did not last in time in the presence of the agonist. Then, the rest of the experiments are describing what occurs in these CBR1-overexpressing neurons in vitro when cytoskeletal elements are being blocked. But, to this reviewer, it is questionable how relevant are the results found in these experiments to the endogenous role of CBR1 in vivo*.

We agree that this is an important question. In order to show the relevance of our *in vitro* results to the endogenous role of CB1Rs, we have now repeated several key experiments with cultured hippocampal neurons overexpressing only the cytoskeleton markers LifeAct-mCherry and EB3-eGFP, but not Flag-CB1R-eCFP, to label dynamic growth cones. We show that in these neurons, the endogenous endocannabinoid 2-AG also induces significant retraction and that the WIN-induced growth cone retraction of neurons, the major *in vitro* experimental read-out of our study, is also dependent of ROCK and NMII. These new results are now included in Figure 1 and the new Figure 3.

*Along the same line, the experiments of in utero electroporating CB1 in cortical neurons are not convincing as the WIN-induced retraction in growth cones expressing only EB3 and LifeAct is very mild and not comparable with the one found in CB1R axons, that incidentally becomes only evident 96 minutes after WIN addition*.

We also recognize in the paper that the effect of WIN is retarded in organotypic slices and we attribute this to retarded tissue diffusion of the highly hydrophobic ligand. The relatively mild averaged effect on axons expressing only EB3 and LifeAct is probably due to the variable level of endogenous CB1R expression in these neurons. We have now added this explanation to the text. This experiment aimed only to show the presence of the NMII-dependent effect of cannabinoids on growth cone progression in a more physiological context. Combined with the long-term *in vivo* data presented on Figure 5, we hope that we can confidently say that actomysin contraction downstream of CB1Rs is important in the development of axonal projections.

*3) Another major concern is the experiments and conclusions made by the authors when blocking NMII or CB1R in vivo by infusion of Blebbistatin or AM251 in the brain of rat embryos (*Figure 4*). The authors state “CB1R-expressing corticofugal axons showed important targeting errors, by invading the subventricular zone, from which CB1R-expressing axons are usually excluded”. Where do they see these errors? First, it is normal to find Tuj1 cut axonal elements in the SVZ in wild type conditions*.

The referees are right: in wild type conditions Tuj1 positive axons also are present in the SVZ but the majority of Tuj1 positive axons are running through in the adjacent intermediate zone (IZ). Our results suggest that both CB1R activation and NMII contractility is important to efficiently segregate corticofugal axons between the IZ and the SVZ; in fact, their pharmacological inhibition induces an increase in the number of Tuj1-positive axons in the SVZ.

*Second*, *how do they know that these Tuj1 axons are corticofugal?*

In order to directly show the presence of corticofugal axons in the SVZ, we have performed a new *in utero* electroporation experiment to label cortical neurons with GFP. We now show on Figure 5—figure supplement 1 the presence of GFP expressing Tuj1-positive axonal elements in the SVZ, suggesting the corticofugal origin of the axons analyzed in our experiment.

*How do they know that these Tuj1 axons express CB1R? From the data provided in this figure there is no evidence of targeting errors of corticofugal axons by blocking CBR1*.

Thank you for the suggestion. Now we directly show an example of co-localization of Tuj1 and CB1R in axons invading the SVZ on Figure 5.

*4) The authors use Neuro2A cells to investigate the role of CBR1 in other neuronal sub-compartments such as the cell soma. To this reviewer, this is not a clear step forward. Why do the authors choose to perform these experiments in a cell line instead of using the hippocampal neurons that they have been using so far? And more importantly*, *what is the relevance of their findings in this regard to the role of CB1R in neuronal development in vivo?*

We chose this platform to show that cannabinoid-induced actomysin contraction is not restricted to the highly-plastic growth cone, but can be detected in other cell domains that contain F-actin. Such a prominent region is the sub-membrane actomyosin cortex that was already well-characterized in non-neuronal cells (46; 45; 5; 9). In order to reproducibly obtain quantitative AFM data on the rigidity of the cell body and to be able to show convincingly on 3D reconstruction the rapid and temporary blebbing of the plasma membrane, reported in cells rounding up for mitosis by using actomysoin contraction, we chose to use N2A neuroblastoma cells for their simpler morphology. In addition, this model also allowed showing the short-term effects of CB1R-induced actomyosin contraction in outgrowth/retraction of non-differentiated neurites, which is an often used experimental read-out to measure cannabinoid-induced structural effects (Table 2 in [14]). Now we have modified the text to better explain the justification of this model.